Workshop at the 6th Symposium on Advances in Approximate Bayesian Inference (non-archival), 2024 1–42

# Shaving Weights with Occam's Razor:
# Bayesian Sparsification for Neural Networks using the Marginal Likelihood

**Rayen Dhahri**                                          RAYEN.DHAHRI@TUM.DE
*Department of Computer Science, Technical University of Munich*

**Alexander Immer**                                  ALEXANDER.IMMER@INF.ETHZ.CH
*Department of Computer Science, ETH Zürich*
*Max Planck Institute for Intelligent Systems, Tübingen*

**Betrand Charpentier**                                  CHARPENT@IN.TUM.DE
*Department of Computer Science, Technical University of Munich*

**Stephan Günnemann**                                  S.GUENNEMANN@TUM.DE
*Department of Computer Science, Technical University of Munich*

**Vincent Fortuin**                                  VINCENT.FORTUIN@TUM.DE
*Department of Computer Science, Technical University of Munich*
*Helmholtz AI, Munich*

## Abstract

Neural network sparsification is a promising avenue to save computational time and memory costs, especially in an age where many successful AI models are becoming too large to naïvely deploy on consumer hardware. While much work has focused on different weight pruning criteria, the overall *sparsifiability* of the network, i.e., its capacity to be pruned without quality loss, has often been overlooked. We present **Spa**rsifiability via the **M**arginal likelihood (**SpaM**), a pruning framework that highlights the effectiveness of using the Bayesian marginal likelihood in conjunction with sparsity-inducing priors for making neural networks more sparsifiable. Our approach implements an *automatic Occam's razor* that selects the most sparsifiable model that still explains the data well, both for structured and unstructured sparsification. In addition, we demonstrate that the pre-computed posterior Hessian approximation used in the Laplace approximation can be re-used to define a cheap pruning criterion, which outperforms many existing (more expensive) approaches. We demonstrate the effectiveness of our framework, especially at high sparsity levels, across a range of different neural network architectures and datasets.

## 1. Introduction

The availability of large datasets and powerful computing infrastructure has fueled the growth of deep learning, enabling the training of massive and complex neural networks. While achieving breakthroughs like high image recognition accuracy (Russakovsky et al., 2015), high-quality text generation (Radford et al., 2022), and catalyzing performance gains across various domains, this development has amplified the challenge of over-parameterization (LeCun et al., 1990; Frankle and Carbin, 2019) and raised concerns about the increase in model size and number of operations. Despite their high performance, over-parameterized neural networks (NNs) present significant deployment challenges, particularly in hardware-constrained environments (Ray, 2022; Paleyes et al., 2022). The quest for sparser neural

networks, while promising for efficiency and interpretability, faces a crucial hurdle: many trained networks are not *sparsifiable*, i.e., they resist effective pruning, regardless of the chosen pruning criteria (Molchanov et al., 2016; Frankle and Carbin, 2019; Liebenwein et al., 2021). Our work tackles this problem, showing that more *sparsifiable* networks can be achieved through Bayesian model selection using the marginal likelihood (MacKay, 1995; Immer et al., 2021a) and the choice of an adequate prior that will induce such sparsifiability.

By leveraging the marginal likelihood in conjunction with well-chosen priors, we use the *automatic Occam's razor* property (Rasmussen and Ghahramani, 2000) of Bayesian model selection, guiding the training process towards models that are inherently more sparsifiable while still faithfully representing the data. This is achieved by optimizing thousands of prior parameters to adaptively regularize weight magnitudes. We make use of recent advancements in Laplace inference for Bayesian neural networks (BNNs) (Immer et al., 2021a; Daxberger et al., 2021), allowing us to approximate the marginal likelihood (MacKay, 1995) efficiently. Once trained, these BNNs can then be more effectively sparsified using different pruning criteria.

Notably, the pre-computed posterior Hessian from the marginal likelihood training readily translates into a powerful pruning criterion similar to the popular Optimal Brain Damage (OBD; LeCun et al., 1990), which we call *Optimal Posterior Damage* (OPD), that can be cheaply computed in practice and often outperforms existing criteria.

Extensive empirical evaluations demonstrate the strength of our SpaM approach and the derived OPD pruning criterion in both unstructured and structured sparsification tasks across various datasets, architectures, and sparsification scheduling scenarios (online and post-hoc). Moreover, they show that our framework strikes a compelling balance between performance and computational cost.

## 2. Shaving Weights with Occam's Razor

We discuss Marginal Likelihood for Deep Learning and Neural Network Pruning in the dedicated background section (Appendix B) necessary for the understanding of our method as well as present the related work in (Appendix C).

We identify sparsifiable neural networks by automatically regularizing (groups of) parameters to have small magnitudes to facilitate pruning the least important ones, both within a probabilistic framework. Specifically, we utilize priors that regularize parameters in potentially structured ways leading to smaller magnitudes. To optimize the resulting plentitude of regularization hyperparameters, we employ the Bayesian marginal likelihood as a differentiable objective function, effectively implementing a Bayesian variant of Occam's razor that drives irrelevant parameters towards smaller magnitudes. While the regularized networks can be pruned with any method, we ultimately propose to use the computed posterior precision for sparsification to only keep well-determined weights of large magnitude.

### 2.1. Structured Priors for Regularization

To reduce the magnitude of parameters and make them more amenable to pruning, we introduce structured priors and show how to combine them with diagonal and KFAC Laplace approximations. While a scalar prior, corresponding to weight decay, is the most common,

it suggests that all parameters in a neural network are equally relevant and favors uniform magnitude of parameters, which is suboptimal for pruning (Hoefler et al., 2021, Sec. 3.6).

Instead of scalar priors, we regularize parameters with different strengths using layer-, unit-, and parameter-wise priors. Layer-wise priors regularize individual layers differently and have been shown to aid pruning and improve generalization (Aghasi et al., 2017; Gordon et al., 2018; Immer et al., 2021a; Antorán et al., 2022). Unit-wise regularization has been used mostly in traditional statistics, for example for group sparsity (Yuan and Lin, 2006), but recently also for channels or feature dimensions in neural networks (Wen et al., 2016; Scardapane et al., 2017).

We consider different priors in the context of the Laplace approximation for marginal likelihood optimization and pruning: Scalar priors correspond to standard weight decay and are identical for all weights. Layer-wise priors provide a scalar regularizer $\delta_l$ per layer that is stacked into a vector $\boldsymbol{\delta}$ in line with the number of parameters per layer. Parameter-wise priors allow to specify $\boldsymbol{\delta}_p$ for each parameter $p$ individually. We define unit-wise priors so that each unit, which denotes a channel for convolutional and a hidden neuron for fully-connected layers, has a regularization strength for incoming and outgoing weights separately. Thus, a weight $\theta_p$ that connects unit $i$ at layer $l$-1 with unit $j$ in layer $l$ has prior $\mathcal{N}(0, [\delta_{l-1}]_i \cdot [\delta_l]_j)$, that is, each layer $l$ with $M_l$ hidden units has a prior vector $\boldsymbol{\delta}_l \in \mathbb{R}^{M_l}$. A weight is thus regularized more strongly whenever both its in- and output neurons are.

Our different priors are simple to combine additively with a diagonal Hessian approximation for the Laplace approximation (Equation (B.2)) but not with a KFAC structure. For that reason, so far, only scalar or layer-wise priors have been used for KFAC posterior approximations (Daxberger et al., 2021). The main issue is that we want to preserve the Kronecker factors and not factor them due to the resulting memory cost. For scalar or layer-wise priors, this can be achieved by an eigendecomposition of the individual factors

$$\mathbf{A} \otimes \mathbf{G} + \mathbf{I}\delta \stackrel{\text{def}}{=} \mathbf{Q}_A \mathbf{\Lambda}_A \mathbf{Q}^\mathsf{T} \otimes \mathbf{Q}_G \mathbf{\Lambda}_G \mathbf{Q}_G^\mathsf{T} + \mathbf{I}\delta = (\mathbf{Q}_A \otimes \mathbf{Q}_G)(\mathbf{\Lambda}_A \otimes \mathbf{\Lambda}_G + \delta)(\mathbf{Q}_A^\mathsf{T} \otimes \mathbf{Q}_G^\mathsf{T}), \quad (1)$$

which means that the precision only needs to be added to the diagonal eigenvalues and no Kronecker product needs to be calculated for inversion or determinant calculation. To add a diagonal prior precision $\boldsymbol{\delta}_l$ with the KFAC of the $l$th layer, we derive an optimal approximation in the KFAC eigenbasis, so as to maintain the Kronecker-factored structure of the posterior. We present the proposition and proof in Theorem 1 This approach is similar to that of George et al. (2018), who correct KFAC's eigenvalues towards the diagonal Gauss-Newton, but solves the problem of adding a full-rank diagonal instead of a rank-1 outer product to the KFAC eigenbasis.

## 2.2. Learning Regularization with the Marginal Likelihood

To optimize the potentially millions of regularization parameters, for example, arising from a parameter-wise prior, we employ the marginal likelihood as a differentiable objective. Optimizing regularization parameters has the advantage that different (groups of) parameters will be regularized differently and therefore become easier to prune. While it would be intractable to optimize that many regularization parameters using validation-based forms of optimization, the marginal likelihood can be estimated and differentiated during training (Immer et al., 2021a, 2023; Lin et al., 2023).

Automatically determining the relevance of parameter-groups (ARD) is a common approach in Bayesian learning that can lead to sparsity and smaller parameter magnitudes (MacKay et al., 1994; Tipping, 2001) and has been used especially in linear models. The Bayesian marginal likelihood provides an objective that automatically regularizes irrelevant parameter-groups more to lower their magnitude. Therefore, it is said to implement a Bayesian variant of Occam's razor by finding the simplest model explaining the data well (Rasmussen and Ghahramani, 2000).

Mathematically, all the prior parameters $\boldsymbol{\delta}$ constitute the hyperparameters of the model $\mathcal{M}$ in the log marginal likelihood (Equation (B.1)) that we optimize interleaved with the neural network parameters. When optimizing the prior parameters, we use gradient ascent

$$\boldsymbol{\delta}_{t+1} \leftarrow \boldsymbol{\delta}_t + \alpha \nabla_{\boldsymbol{\delta}} \log p(\mathcal{D}|\boldsymbol{\delta})|_{\boldsymbol{\delta}=\boldsymbol{\delta}_t}, \tag{2}$$

or adaptive optimizers like Adam (Kingma and Ba, 2014). Algorithmically, we follow Immer et al. (2021a) and optimize the Laplace approximation to the marginal likelihood after an initial burn-in phase with a certain frequency.

## 2.3. Optimal Posterior Damage (OPD)

While sparsity regularization learned by marginal likelihood training can advantageously be combined with any pruning criteria like Single-shot Network Pruning (SNIP; Lee et al., 2018), variants of Gradient Signal Preservation (GraSP; Wang et al., 2020; Lubana and Dick, 2021; Rachwan et al., 2022), or magnitude pruning (Han et al., 2016), we propose in this section a new pruning criterion which uses our Laplace approximation and extends the unstructured Optimal Brain Damage (OBD) pruning criterion (LeCun et al., 1990). While OBD traditionally uses the prior precision $\mathbf{H}_{\boldsymbol{\theta}}$, we propose to adapt it to use the posterior precision $\mathbf{P}_{\boldsymbol{\theta}}$. In this case the importance score $S(\theta_p)$ for a given parameter becomes

$$S(\theta_p) = \mathrm{P}_{pp} \times \theta_p^2 \tag{3}$$

where $\mathrm{P}_{pp}$ denotes the posterior precision for the parameter $\theta_p$, extracted from the diagonal of the posterior precision matrix $\mathbf{P}_{\boldsymbol{\theta}}$. We call this novel posterior-based pruning criterion *Optimal Posterior Damage* (OPD). Intuitively, individual weights with high scores indicate certainty of the posterior distribution and a significant contribution to the model's functionality, as indicated by the magnitude. In the unstructured process, we perform global one-shot pruning.

Further, we propose a structured version of OPD by aggregating the score of a structured set of parameters $g$, i.e.,

$$S(g) = \sum_{p \in g} S(\theta_p) = \sum_{p \in g} \mathrm{P}_{pp} \times \theta_p^2 \tag{4}$$

In practice, the structured set of parameters $g$ corresponds to all parameters along one dimension of the weight matrix inside a layer, in order to reduce the size of the matrix multiplications. Since subsequent layers might have significantly different weight matrix dimensions impacting the magnitude of the aggregated sum, we opt for uniform structured pruning to guarantee a fair pruning treatment across all layers. Moreover, as removing a full structure is more aggressive, we also apply gradual pruning during training. Finally,

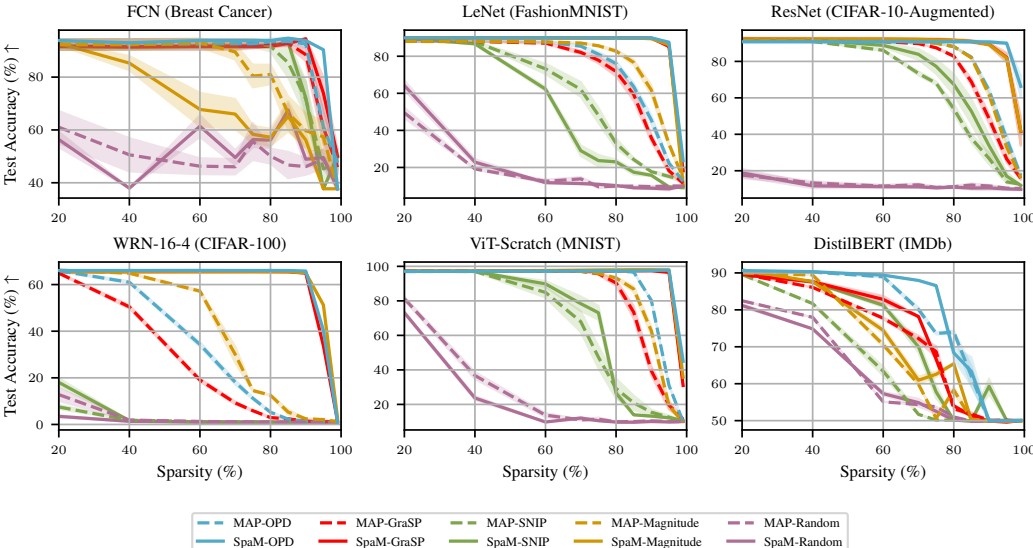

Figure 1: Predictive performance as a function of sparsity level in unstructured pruning. We see that SpaM improves the performance over MAP training across most architectures, datasets, and pruning criteria, and that OPD often outperforms the other pruning criteria. Both of these effects are particularly visible at higher sparsity levels.

we omit pruning the final layer to mitigate overly strong impacts on classification accuracy and computational stability (Elsen et al., 2019).

The OPD pruning criterion can be computed to prune *online* during training or *post-hoc* after training by using Laplace approximations. On one hand, in the case of maximum a posteriori (MAP) training, Laplace approximations of the inverse Hessian at $\boldsymbol{\theta}_*$ can be additionally computed to approximate OPD. On the other hand, in the case of our marginal likelihood training, Laplace approximations of the precision matrix can be reused to compute OPD without computational overhead, in contrast to the other pruning criteria, which often require additional computations to be performed. Note that we will also show in our experiments that OPD additionally avoids the need for potentially expensive fine-tuning after pruning.

## 3. Experiments

We conduct experiments on various datasets and models and outline our experimental setup in detail in Appendix G. We compare MAP training with our proposed SpaM approach with different priors, comparing our OPD pruning criterion with random pruning, magnitude pruning, SNIP (Lee et al., 2018), GraSP (Wang et al., 2020; Lubana and Dick, 2021), and SynFlow (Tanaka et al., 2020). We show that **SpaM improves pruning performance across a range of different pruning criteria**, especially at higher sparsities, and that our **OPD often outperforms the baselines**. This observation extends not only to predictive accuracy, but also uncertainty estimation. Moreover, we show that the choice of prior can play a significant role and **introduce parameter-wise and unit-wise priors** for the KFAC approximation. Finally, we show that SpaM and OPD also work in a structured pruning setting, leading to **significant computational benefits**.

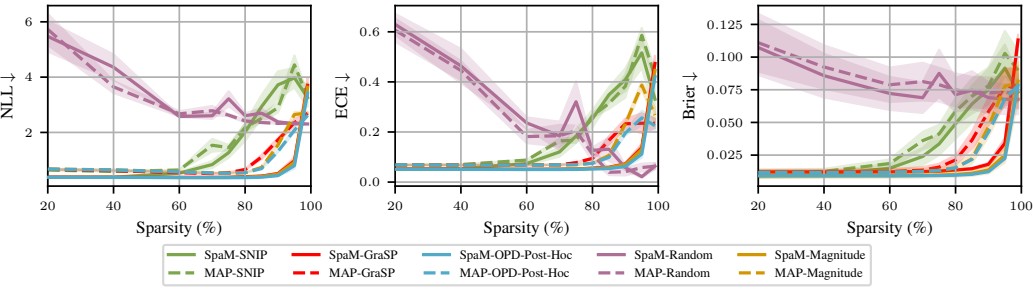

Figure 2: Uncertainty estimation with pruned ResNets on CIFAR-10. We see that SpaM improves uncertainty estimation in terms of NLL, ECE, and Brier score for many pruning criteria and that our OPD criterion outperforms the baselines, especially at high sparsities.

**SpaM Improves Performance at High Sparsities.** We compare SpaM to MAP training with different pruning criteria, including OPD, across different models and datasets. For SpaM in this unstructured pruning context, we use the diagonal Laplace approximation with a parameter-wise prior. Encouragingly, MAP and SpaM reach comparable performance during training, showing that the increased sparsifiability of SpaM comes at no additional cost in unpruned performance (see Figure E.1 in the appendix).

More excitingly, we see in Figure 1 and Table C.1 that SpaM drastically improves the performance for many pruning criteria, especially magnitude pruning, GraSP, and OPD. We also see that OPD, despite being a cheap byproduct of our marginal likelihood computation, often outperforms the other pruning criteria, especially at higher sparsities. For instance, at 95 % pruning rate (i.e., with 20x fewer parameters), our combination of SpaM and OPD still retains almost the same performance as the unpruned model, while the other pruning criteria with MAP training have dropped to unusable performance levels at this sparsity.

**Fine-tuning.** We see in Figure E.9 in the appendix that some of this performance difference can be remedied by costly fine-tuning of the networks after pruning, which however still does not allow the other methods to reach the full SpaM-OPD performance, and in the case of OPD, also does not further improve its already near-optimal performance.

**Online pruning.** Figure E.8 shows that our online version of SpaM, which uses the marginal likelihood and OPD during training to iteratively prune the network, reaches comparable performance levels to the post-hoc version, thus offering a computationally even more convenient way to effectively sparsify neural networks.

**Uncertainty estimation.** Given that SpaM is a Bayesian method, it does not only offer high predictive accuracies but also calibrated uncertainty estimates. Indeed, we see in Figure 2 that the trends we have seen for accuracy also apply for negative log-likelihood, expected calibration error, and the Brier score. Again, SpaM improves the uncertainty estimates over MAP training, OPD outperforms most other criteria, and we achieve well-calibrated models up until very high sparsity levels. Note that the random baseline also achieves a low ECE at high sparsity levels because it essentially reverts to random guessing, which is a known weakness of the ECE metric (Gruber and Buettner, 2022).

We discuss the impact of prior selection on OPD and other methods in Appendix E.2, SpaM's extension to structured sparsity Appendix G.6, and results on modern architectures Appendix E.9.

## 4. Conclusion

We leverage the Bayesian marginal likelihood to identify inherently sparsifiable neural networks, achieving significant performance and uncertainty estimation gains at high sparsity levels across diverse pruning criteria. Notably, our novel pruning criterion, OPD, efficiently leverages the marginal likelihood computation. Additionally, we present guidelines for effective prior selection and demonstrate the efficiency of our SpaM approach for structured pruning. This work paves the way for deploying large, pruned AI models on resource-constrained devices.

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

# Appendix A. Method Overview

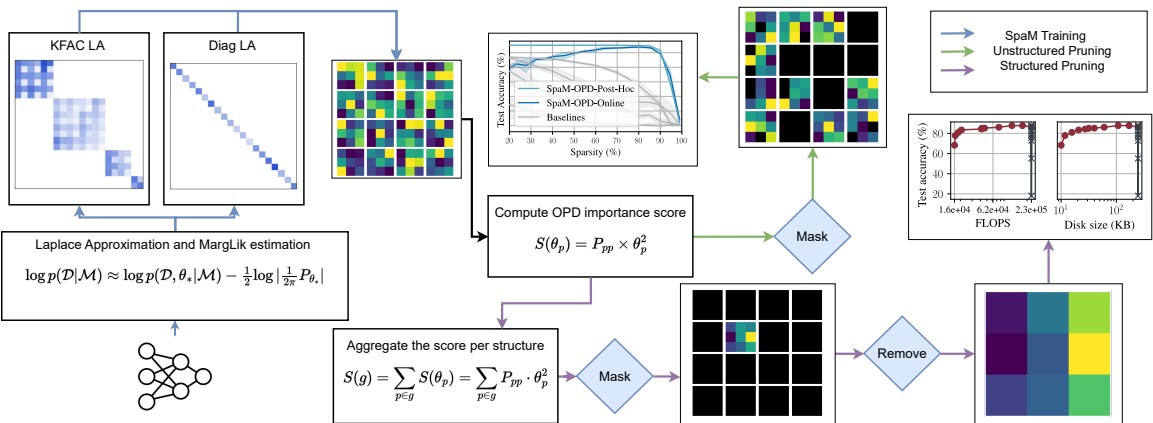

Figure A.1: Overview of our proposed SpaM method. We start by training the network to maximize the marginal likelihood using the Laplace approximation, while simplifying the Hessian computation through either the KFAC or diagonal approximation. We can then use our precomputed posterior Hessian as a pruning criterion (OPD). For the unstructured case, we compute thresholds to achieve different target sparsities, compute the mask, and apply it, while for the structured approach, we aggregate the score per layer for easier weight transfer, compute the mask, and then delete the masked structures to obtain a smaller model.

# Appendix B. Background

We use deep neural networks to model learning tasks with inputs $\mathbf{x}_n \in \mathbb{R}^D$ and targets $\mathbf{y}_n \in \mathbb{R}^C$ collected in a dataset $\mathcal{D} = \{(\mathbf{x}_n, \mathbf{y}_n)\}_{n=1}^N$ of $N$ pairs. A model is parameterized by weights $\boldsymbol{\theta} \in \mathbb{R}^P$, and maps from inputs to targets using the neural network function $\mathbf{f}_{\boldsymbol{\theta}}(\mathbf{x})$. Assuming the data are *i.i.d.*, we have a likelihood $p(\mathcal{D}|\boldsymbol{\theta}) = \prod_{n=1}^N p(\mathbf{y}_n|\mathbf{f}_{\boldsymbol{\theta}}(\mathbf{x}_n))$. We minimize the negative log likelihood, which corresponds to common losses like the cross-entropy in classification. Additionally, regularization in the form of weight decay is commonly used and corresponds to a Gaussian prior on parameters $p(\boldsymbol{\theta}) = \mathcal{N}(\boldsymbol{\theta}; \mathbf{0}, \text{diag}(\boldsymbol{\delta}))$ with diagonal precision.

## B.1. Marginal Likelihood for Deep Learning

**The marginal likelihood** serves as the probabilistic foundation for model evaluation and selection. It provides an objective to optimize the tradeoff between data fit and model complexity, akin to the concept of Occam's razor (Rasmussen and Ghahramani, 2000; MacKay, 2002), by quantifying how well a model $\mathcal{M}$, with all its inherent uncertainties, explains the observed data:

$$p(\mathcal{D}|\mathcal{M}) = \int p(\mathcal{D}|\boldsymbol{\theta}, \mathcal{M}) \, p(\boldsymbol{\theta}|\mathcal{M}) \, \mathrm{d}\boldsymbol{\theta}. \tag{B.1}$$

However, it requires computing an intractable integral over all neural network parameters.

**The Laplace approximation** (LA, MacKay, 1992) provides a tractable and effective approximation to the marginal likelihood for deep learning (Immer et al., 2021a). It arises from a second-order Taylor approximation around an estimate of the mode, $\boldsymbol{\theta}_*$, resulting in

$$\log p(\mathcal{D}|\mathcal{M}) \approx \log p(\mathcal{D}, \boldsymbol{\theta}_*|\mathcal{M}) - \tfrac{1}{2}\log|\tfrac{1}{2\pi}\mathbf{P}_{\boldsymbol{\theta}_*}|, \tag{B.2}$$

where $\mathbf{P}_{\boldsymbol{\theta}_*}$ is the posterior precision given by the Hessian of the negative log joint distribution, $-\nabla_{\boldsymbol{\theta}}^2 \log p(\mathcal{D}, \boldsymbol{\theta}|\mathcal{M})$, evaluated at $\boldsymbol{\theta}_*$. Defining $\mathbf{H}_{\boldsymbol{\theta}_*}$ as the Hessian of the negative log likelihood objective $-\nabla_{\boldsymbol{\theta}}^2 \log p(\mathcal{D}|\boldsymbol{\theta}, \mathcal{M})$, the posterior precision decomposes as $\mathbf{P}_{\boldsymbol{\theta}_*} = \mathbf{H}_{\boldsymbol{\theta}_*} + \mathrm{diag}(\boldsymbol{\delta})$.

In practice, the Hessian of the negative log likelihood is often approximated by the positive semidefinite **generalized Gauss-Newton** (GGN, Schraudolph, 2002),

$$\mathbf{H}_{\boldsymbol{\theta}} \approx \sum_{n=1}^N \nabla_{\boldsymbol{\theta}}\mathbf{f}_{\boldsymbol{\theta}}(\mathbf{x}_n)\nabla_{\mathbf{f}}^2 \log p(\mathbf{y}_n|\mathbf{f}_{\boldsymbol{\theta}}(\mathbf{x}_n))\nabla_{\boldsymbol{\theta}}^{\mathsf{T}}\mathbf{f}_{\boldsymbol{\theta}}(\mathbf{x}_n), \tag{B.3}$$

which relies on the Jacobians of the neural network function and second derivative of the negative log likelihood at the output. Further, it is amenable to efficient structured approximations like diagonal or layer-wise variants (e.g., Martens and Grosse, 2015; Botev et al., 2017).

**Diagonal and block-diagonal GGN approximations** are efficient and therefore commonly used for Laplace approximations in deep learning (Ritter et al., 2018; Daxberger et al., 2021). The diagonal LA is cheap in terms of storage and computation by only modeling the marginal variances of parameters. Kronecker-factored LA (KFAC LA, Ritter et al., 2018) instead relies on a block-diagonal approximation to the GGN of the parameters $\boldsymbol{\theta}_l$ in the $l$th layer,

$$\mathbf{H}_{\boldsymbol{\theta}_l} \approx \mathbf{A}_l \otimes \mathbf{G}_l, \tag{B.4}$$

where the factors are given by the outer products of pre-activations and Jacobians w.r.t. the output of a layer, respectively (Martens and Grosse, 2015; Botev et al., 2017). The top left of Figure A.1 shows a comparison of both structures.

## B.2. Neural Network Pruning

**Unstructured and structured pruning.** The goal of the pruning procedure is to remove parameters from $\boldsymbol{\theta}$ without affecting the quality of the model output $\mathbf{f}_{\boldsymbol{\theta}}(\mathbf{x})$. While unstructured pruning consists in zeroing individual entries $\theta_p$ of the weight matrices, structured pruning consists in deleting entire structured sets of parameters $g$, like rows or columns (Liang et al., 2021; Fang et al., 2023). The results of structured pruning enable smaller matrix multiplications which directly provide real-world efficiency gains on most hardware, including GPUs.

Pruning procedures usually follow three steps: **(1)** We use a scoring function $S(\cdot)$ to evaluate the importance of each individual parameter $S(\theta_p)$ for unstructured pruning, or structured set of parameters $S(g)$ for structured pruning. **(2)** We compute a binary mask $\mathbf{m}$ with the same dimensions as $\boldsymbol{\theta}$ which assign 0 values to parameters whose unstructured or structured pruning scores are below a threshold $T$, and 1 otherwise. While the threshold $T$ is determined based on the target sparsity across layers for *global* pruning, it is determined

Table C.1: Accuracies of pruned ResNets on CIFAR-10. The best training method for each pruning criterion is highlighted in green, where we see that SpaM improves performance for all criteria except the random baseline. The best performances overall at each sparsity level are shown in **bold**, showing that our OPD criterion outperforms the others at high sparsities.

| Criterion | Training | Sparsity (%) | | | | |
|---|---|---|---|---|---|---|
| | | 80 | 85 | 90 | 95 | 99 |
| OPD | MAP | 88.06 (±0.12) | 82.32 (±0.44) | 64.08 (±1.32) | 37.52 (±2.34) | 17.32 (±1.01) |
| | SpaM | 90.78 (±0.66) | 90.78 (±0.65) | **90.68 (±0.65)** | **89.98 (±0.61)** | **66.28 (±5.89)** |
| GraSP | MAP | 82.87 (±0.48) | 68.78 (±1.88) | 48.65 (±2.69) | 26.46 (±1.86) | 15.75 (±0.80) |
| | SpaM | 91.50 (±0.66) | **90.94 (±0.65)** | 89.42 (±0.71) | 82.18 (±2.65) | 41.48 (±7.95) |
| SNIP | MAP | 53.96 (±2.72) | 37.74 (±2.21) | 26.74 (±3.17) | 13.88 (±0.87) | 12.58 (±0.36) |
| | SpaM | 67.40 (±5.68) | 52.62 (±6.84) | 33.75 (±5.71) | 17.06 (±2.23) | 11.90 (±0.51) |
| Magnitude | MAP | 88.17 (±0.12) | 81.92 (±0.37) | 61.60 (±1.11) | 32.88 (±1.52) | 16.12 (±0.90) |
| | SpaM | **91.55 (±0.64)** | 90.92 (±0.64) | 89.23 (±0.62) | 81.80 (±2.22) | 41.78 (±7.20) |
| Random | MAP | 11.25 (±0.48) | 12.15 (±0.92) | 11.65 (±0.62) | 10.45 (±0.17) | 10.27 (±0.17) |
| | SpaM | 11.00 (±0.48) | 10.47 (±0.86) | 10.56 (±1.15) | 10.01 (±0.45) | 9.81 (±0.61) |

per layer for *uniform* pruning (Liang et al., 2021). **(3)** We apply the mask on the weight matrix with element-wise multiplication $\mathbf{m} \circ \boldsymbol{\theta}$ to effectively remove the least important parameters. Alternatively, structured pruning enables to directly remove rows or columns whose mask values are 0 to reduce weight matrix dimensions.

**Pruning before, during, after training.** The pruning procedure can be applied at different times. On the one hand, some methods prune *before* or *during* training (Lee et al., 2018; Rachwan et al., 2022), thus allowing to (partially) train sparse models. On the other hand, other methods propose to prune *after* training (LeCun et al., 1990), which allows the compression of existing pre-trained models.

**One-shot and iterative pruning.** While the pruning procedure can be applied a single time to reach the target sparsity in *one-shot* manner, it is also possible to apply it *iteratively* to reach the target sparsity with smaller steps. Further, the pruning iterations can also be spaced at a specified pruning interval during training to *gradually* compress the model.

## Appendix C. Related work

**Laplace-approximated BNNs.** From the early inception of Bayesian neural networks (Neal, 1992; Hinton and Van Camp, 1993), the Laplace approximation was a popular inference method (MacKay, 1992). In recent years, it has undergone a renaissance (Martens and Grosse, 2015; Botev et al., 2017; Ritter et al., 2018; Daxberger et al., 2021), including critical work on using more scalable approximations for the associated marginal likelihood in the context of model selection (MacKay, 1995; Immer et al., 2021a, 2022), which we use in our framework. To the best of our knowledge, we are the first to study the benefits of this Laplace-approximated marginal likelihood in the context of sparsification of deep neural networks. However, similar methods that automatically quantify the relevance (ARD) of

parameters have been derived and used for linear and kernel models (Tipping, 2001; Wipf and Rao, 2004).

**Pruning neural networks.** Various pruning criteria have been proposed to determine the importance of model parameters. Many criteria proposed to prune based on the weight magnitude (Han et al., 2016; Zhou et al., 2020; Bellec et al., 2018) but usually required additional fine-tuning to recover accuracy. Sun et al. (2023) proposed to combine activation and weight norms for pruning without fine-tuning. Other approaches include pruning using first-order information based on connectivity (Lee et al., 2018) or synaptic flow conservation (Tanaka et al., 2020), or second-order information aiming at preserving gradient flow (Rachwan et al., 2022; Wang et al., 2020; Lubana and Dick, 2021). Recently, van der Ouderaa et al. (2023) focused on pruning LLMs based on a second-order Taylor expansion. In contrast, OPD uses second-order information provided by the posterior precision given by the Laplace approximation.

Beyond pruning criteria, there have been many approaches to prune at initialization (Lee et al., 2018; Wang et al., 2020; Tanaka et al., 2020), during training (Golkar et al., 2019; Zhou et al., 2021), and after training (Han et al., 2016; Sun et al., 2023). In particular, multiple works proposed to leverage specific training schemes promoting zero-invariant parameter groups for structured pruning (Chen et al., 2021, 2023). In contrast, SpaM induces sparse structures during training using Bayesian marginal likelihood training.

# Appendix D. Proof for Diagonal Prior in a Kronecker-factored Eigenbasis

**Proposition 1 (Diagonal Prior in KFAC Eigenbasis)** *Considering the Frobenius norm, the optimal diagonal perturbation of the KFAC eigenvalues $\boldsymbol{\Lambda}_A \otimes \boldsymbol{\Lambda}_B$ to add a diagonal prior precision is given by $\boldsymbol{\Lambda}_A \otimes \boldsymbol{\Lambda}_B + \hat{\boldsymbol{\delta}}$ with $\mathrm{mat}(\hat{\boldsymbol{\delta}}) = (\mathbf{Q}_G^\intercal)^2 \mathrm{mat}(\boldsymbol{\delta}) \mathbf{Q}_A^2$ where the square is element-wise and $\mathrm{mat}(\cdot)$ reshapes the vector to match the parameter shape used in KFAC. Thus, it can be computed efficiently without computing a Kronecker product.*

**Proof** We prove this result in two steps. First, we show what the optimum looks like in terms of the Frobenius norm. Second, we show how to simplify the results to enable efficient computation without computing Kronecker products. We have a KFAC Hessian approximation $\mathbf{A} \otimes \mathbf{B}$ with $\mathbf{A} \in \mathbb{R}^{D_{\mathrm{in}} \times D_{\mathrm{in}}}$ and $\mathbf{B} \in \mathbb{R}^{D_{\mathrm{out}} \times D_{\mathrm{out}}}$ where the dimensionalities $D$. depend on the layer type (Martens and Grosse, 2015). In the case of a fully-connected layer, these are simply the dimensionality of the in- and output hidden representation. The same layer will have $D_{\mathrm{in}} \times D_{\mathrm{out}}$ parameters and thus the corresponding diagonal prior precision is given by $\boldsymbol{\delta} \in \mathbb{R}^{D_{\mathrm{in}} D_{\mathrm{out}}}$. For the Laplace approximation, the eigendecomposition of individual Kronecker factors is already computed as $\mathbf{A} = \mathbf{Q}_A \boldsymbol{\Lambda}_A \mathbf{Q}_A^\intercal$ and similarly for $\mathbf{G}$ as shown in Equation (1). Recall also that $\mathrm{diag}(\cdot)$ turns a vector into a diagonal matrix and extracts the diagonal entries of a matrix into a vector. We are interested in the Frobenius-optimal diagonal perturbation of the eigenvalues so as to maintain the efficiency structure of the

KFAC and thus the downstream Laplace approximation:

$$\underset{\hat{\boldsymbol{\delta}}}{\arg\min} \, \|(\mathbf{Q}_A \otimes \mathbf{Q}_G)(\boldsymbol{\Lambda}_A \otimes \boldsymbol{\Lambda}_G + \operatorname{diag}(\hat{\boldsymbol{\delta}}))(\mathbf{Q}_A^\mathsf{T} \otimes \mathbf{Q}_G^\mathsf{T})$$

$$- (\mathbf{Q}_A \otimes \mathbf{Q}_G)(\boldsymbol{\Lambda}_A \otimes \boldsymbol{\Lambda}_G)(\mathbf{Q}_A^\mathsf{T} \otimes \mathbf{Q}_G^\mathsf{T}) + \operatorname{diag}(\boldsymbol{\delta})\|_F^2$$

$$= \underset{\hat{\boldsymbol{\delta}}}{\arg\min} \, \|\boldsymbol{\Lambda}_A \otimes \boldsymbol{\Lambda}_G + \operatorname{diag}(\hat{\boldsymbol{\delta}}) - \boldsymbol{\Lambda}_A \otimes \boldsymbol{\Lambda}_G + (\mathbf{Q}_A^\mathsf{T} \otimes \mathbf{Q}_G^\mathsf{T}) \operatorname{diag}(\boldsymbol{\delta})(\mathbf{Q}_A \otimes \mathbf{Q}_G)\|_F^2$$

$$= \operatorname{diag}((\mathbf{Q}_A^\mathsf{T} \otimes \mathbf{Q}_G^\mathsf{T}) \operatorname{diag}(\boldsymbol{\delta})(\mathbf{Q}_A \otimes \mathbf{Q}_G)),$$

where we first multiplied the orthogonal bases from left and right and then realized that the values of $\hat{\boldsymbol{\delta}}$ need to be set to the entries of the prior $\boldsymbol{\delta}$ projected into the basis.

Naïvely, computing the optimum of $\hat{\boldsymbol{\delta}}$ would require expanding the Kronecker product above and lead to a potentially intractable complexity of $\mathcal{O}(D_\text{in}^2 D_\text{out}^2)$. However, it is possible to simplify it further to maintain efficient computation: For simplicity, consider the case without Kronecker factorization. We have

$$\operatorname{diag}(\mathbf{Q}^\mathsf{T} \operatorname{diag}(\mathbf{d})\mathbf{Q}) = (\mathbf{Q}^\mathsf{T} \circ \mathbf{Q}^\mathsf{T})\mathbf{d},$$

where $\circ$ is the element-wise Hadamard product. So we can express the diagonal of the matrix-matrix product as a matrix-vector product with the diagonal $\mathbf{d}$ as the vector. In the Kronecker-factored case, we need just one more simplification:

$$\operatorname{diag}((\mathbf{Q}_A^\mathsf{T} \otimes \mathbf{Q}_G^\mathsf{T}) \operatorname{diag}(\boldsymbol{\delta})(\mathbf{Q}_A \otimes \mathbf{Q}_G)) = ((\mathbf{Q}_A^\mathsf{T} \otimes \mathbf{Q}_G^\mathsf{T}) \circ (\mathbf{Q}_A^\mathsf{T} \otimes \mathbf{Q}_G^\mathsf{T}))\boldsymbol{\delta}$$

$$= ((\mathbf{Q}_A^\mathsf{T} \circ \mathbf{Q}_A^\mathsf{T}) \otimes (\mathbf{Q}_G^\mathsf{T} \circ \mathbf{Q}_G^\mathsf{T}))\boldsymbol{\delta}$$

$$= \operatorname{vec}((\mathbf{Q}_G^\mathsf{T} \circ \mathbf{Q}_G^\mathsf{T}) \operatorname{mat}(\boldsymbol{\delta})(\mathbf{Q}_A \circ \mathbf{Q}_A))$$

$$= \operatorname{vec}(\mathbf{Q}_G^\mathsf{T})^2 \operatorname{mat}(\boldsymbol{\delta})\mathbf{Q}_A^2,$$

where we have used the mixed-product property of the Kronecker product and the properties for multiplying a Kronecker-product with a vector. The vec operator "flattens" a matrix, that is, turns a $D_\text{out} \times D_\text{in}$ matrix into a $D_\text{out}D_\text{in}$ vector, and mat does the opposite. The final approximation $\hat{\boldsymbol{\delta}}$ can be computed efficiently in $\mathcal{O}(D_\text{in}^2 + D_\text{out}^2)$. ∎

## Appendix E. Additional Results

### E.1. Baselines

Figure E.1 illustrates that both MAP and SPAM achieve similar levels of performance throughout the training process. This observation underscores that SPAM's enhanced sparsifiability is achieved without compromising the unpruned performance. Furthermore, the comparable unpruned accuracies of SPAM and MAP models indicate that SPAM's sparsifiability benefits are not merely a result of higher baseline accuracies, but rather a distinct advantage offered by the SPAM methodology. The sparsification methods are performed on these models in a way that once the model is trained for a specific seed, we copy it and use it to perform the different sparsification methods, we repeat the steps for a minimum of 4 different seeds ensuring the robustness of our findings.

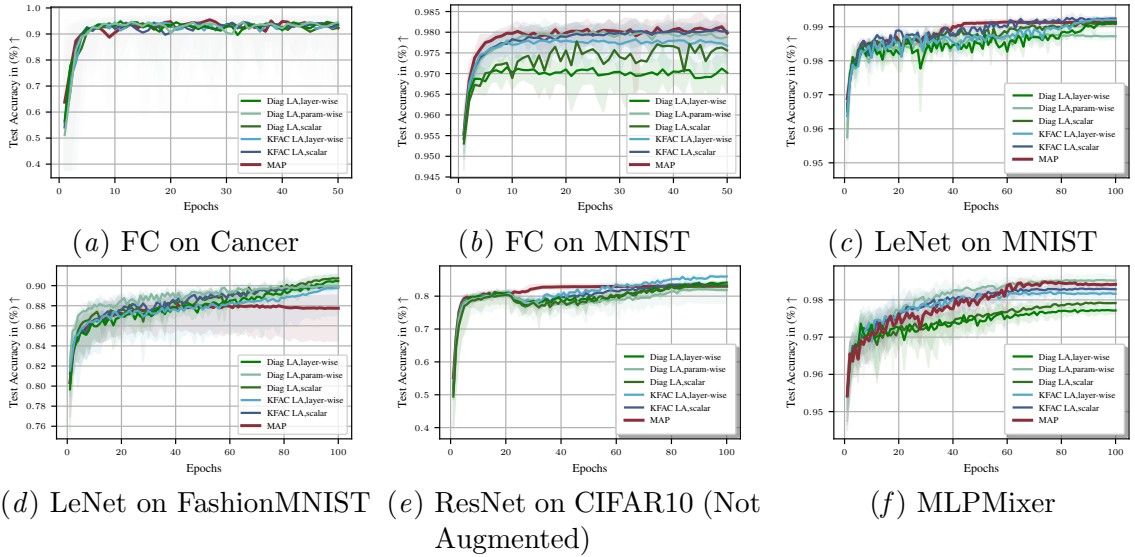

Figure E.1: Training curves for MAP and SpaM training with different priors and Hessian approximations. We see that all methods achieve a similar performance by the end of training.

## E.2. Influence of Priors on Sparsifiability

The Bayesian marginal likelihood, as employed in SpaM, strongly depends on the chosen prior. To understand the influence of prior and Hessian approximation on performance in our proposed SpaM-OPD approach, we compare diagonal and KFAC approximations with scalar, layer-wise, unit-wise, and parameter-wise priors. Note regarding the latter two, that in this work, we are the first to ever implement them for the KFAC approximation, thus contributing to the general framework of Laplace-approximated BNNs (Daxberger et al., 2021), independent of the pruning use case.

We see in Figure E.2 that our newly introduced unit-wise and parameter-wise priors for KFAC indeed outperform the others, especially at high sparsities. When comparing KFAC to the diagonal approximation, we see that KFAC often leads to slightly better performance at lower sparsity levels. However, we also see that the relatively simple choice of parameter-wise prior and diagonal Hessian approximation, as used in our previous experiments above, is a strong baseline across the board and can be recommended as a safe default option for unstructured pruning. Note that the unit-wise priors can be especially useful for structured pruning, as we will see in the following experiment. More detailed prior comparisons can be found in Appendix E.5.

## E.3. SpaM Extends to Structured Sparsification

Here, we study the effect of SpaM and OPD in the more challenging setting of eliminating entire network structures, such as convolutional kernels. Studying different network architectures, we aim to generalize our unstructured pruning approach to the setting of structured pruning, where the structures can be freely defined depending on the use case.

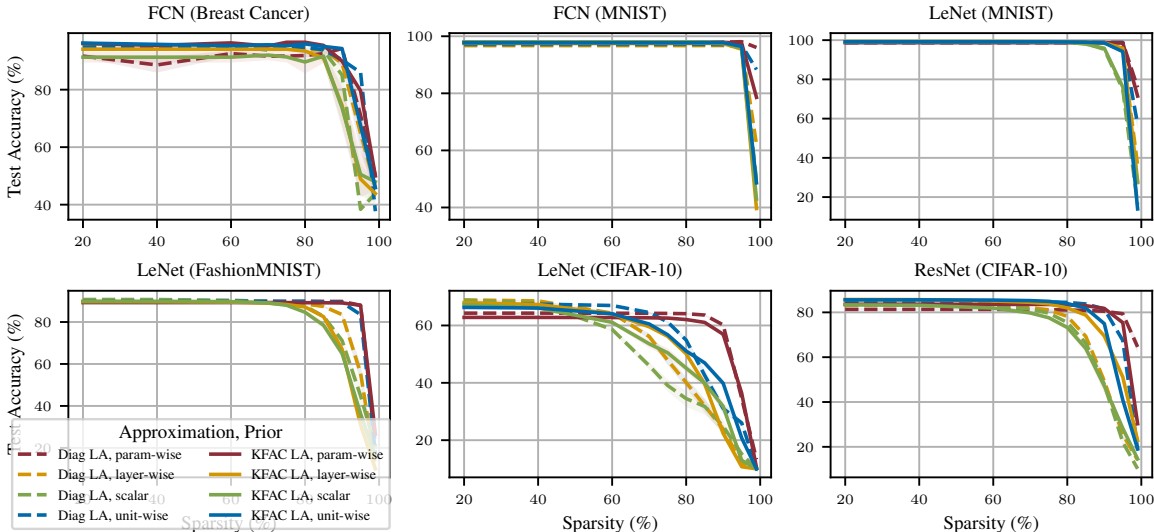

Figure E.2: Comparison of different priors and Hessian approximations for SpaM-OPD pruning. The unit-wise and parameter-wise priors show better performance at high sparsity levels, with the parameter-wise one bridging the gap between Diag and KFAC LA.

Encouragingly, we see in Figure E.3 that our findings from the unstructured case transfer qualitatively also to the structured case, with SpaM-OPD outperforming the baselines at high sparsities. Crucially, while the sparsity patterns generated by unstructured pruning are more difficult to translate into computational benefits, structured pruning directly leads to computational savings on standard GPUs (see also Figure E.18 in the appendix). We see in Figure E.4 that SpaM-OPD dominates the Pareto frontier of the tradeoff between performance and computational cost at high sparsities (i.e., low costs), yielding 10x–20x savings in FLOPS and memory consumption with only minimal deterioration in performance. This positions our proposed framework as a potentially promising approach for the deployment of AI models in resource-constrained environments.

### E.4. Tables

In tables E.1 and E.2, we present our results comparing different methods using MAP and SpaM with various priors. Notably, SpaM with Diag LA and parameter-wise priors significantly outperforms MAP and other SpaM variants at high sparsity levels.

### E.5. Prior effects

Figure E.5 and Figure E.6 illustrate our findings when applying SpaM with various priors for both OPD and GraSP. Notably, Diag LA, using parameter-wise priors, excels in high sparsity scenarios, even with complex models and datasets like ResNets. Furthermore, for MLPmixer, we observe that SpaM variants, employing parameter-wise priors and layerwise approaches, preserve baseline accuracy even at extreme sparsities of 99%.

| Criterion | Approximation | Sparsity (%) Prior | 20 | 40 | 60 | 70 | 75 | 80 | 85 | 90 | 95 | 99 |
|---|---|---|---|---|---|---|---|---|---|---|---|---|
| GraSP | Diag | parameter-wise | 98.75 | 98.75 | 98.74 | 98.75 | 98.75 | 98.75 | 98.75 | **98.74** | **98.72** | 52.19 |
| | | layerwise | 99.08 | 99.09 | 99.10 | 98.96 | 98.50 | 97.79 | 93.50 | 78.93 | 63.71 | 16.67 |
| | | scalar | 99.11 | 99.11 | 99.12 | 98.84 | 98.41 | 97.74 | 87.67 | 57.99 | 44.60 | 15.73 |
| | KFAC | layerwise | **99.25** | **99.25** | **99.26** | 99.01 | 98.74 | 97.86 | 91.48 | 82.78 | 50.75 | 13.48 |
| | | scalar | 99.24 | 99.24 | 99.24 | 99.02 | 98.58 | 98.02 | 92.03 | 73.64 | 42.18 | 12.23 |
| | MAP | MAP | 99.01 | 99.01 | 99.01 | 98.99 | 98.91 | 98.75 | 98.28 | 96.10 | 77.30 | 20.43 |
| SNIP | Diag | parameter-wise | 98.73 | 98.73 | 98.68 | 97.95 | 95.70 | 83.90 | 57.55 | 20.47 | 13.00 | 9.10 |
| | | layerwise | 99.08 | 98.34 | 46.42 | 17.69 | 21.89 | 14.61 | 14.47 | 13.50 | 11.94 | 9.83 |
| | | scalar | 99.11 | 99.11 | 98.56 | 91.82 | 84.70 | 62.25 | 34.56 | 10.75 | 16.28 | 16.70 |
| | KFAC | layerwise | **99.25** | 99.20 | 79.59 | 27.70 | 43.76 | 23.85 | 10.28 | 16.19 | 19.90 | 10.27 |
| | | scalar | 99.24 | 99.24 | 98.56 | 94.82 | 70.25 | 48.28 | 27.02 | 26.88 | 25.95 | 9.86 |
| | MAP | MAP | 99.01 | 99.01 | 98.95 | 98.31 | 97.21 | 94.26 | 87.19 | 65.42 | 25.15 | 12.56 |
| OPD | Diag | parameter-wise | 98.72 | 98.72 | 98.72 | 98.72 | 98.72 | 98.72 | 98.72 | 98.72 | **98.72** | **75.92** |
| | | layerwise | 99.08 | 99.09 | 99.08 | 99.07 | 99.04 | 98.98 | 98.84 | 98.40 | 94.71 | 36.25 |
| | | scalar | 99.11 | 99.11 | 99.10 | 99.11 | 99.06 | 98.89 | 98.28 | 95.32 | 74.78 | 16.12 |
| | KFAC | layerwise | **99.25** | **99.25** | 99.24 | **99.23** | **99.18** | **99.11** | **98.95** | 98.60 | 95.90 | 28.16 |
| | | scalar | 99.24 | 99.24 | 99.24 | 99.16 | 99.10 | 98.90 | 97.97 | 95.88 | 76.11 | 27.38 |
| | MAP | MAP | 99.01 | 99.01 | 99.03 | 98.99 | 98.95 | 98.90 | 98.71 | 98.17 | 92.82 | 27.19 |
| Magnitude | Diag | parameter-wise | 98.72 | 98.72 | 98.72 | 98.70 | 98.69 | 98.67 | 98.65 | 98.59 | 98.03 | 38.61 |
| | | layerwise | 99.08 | 99.09 | 99.08 | 99.06 | 99.01 | 98.92 | 98.46 | 94.20 | 39.86 | 10.19 |
| | | scalar | 99.11 | 99.11 | 99.09 | 99.12 | 99.07 | 98.98 | 98.48 | 95.69 | 74.30 | 13.53 |
| | KFAC | layerwise | **99.25** | **99.25** | 99.22 | 99.18 | 99.08 | 98.95 | 98.38 | 91.93 | 28.52 | 9.80 |
| | | scalar | 99.24 | 99.24 | 99.20 | 99.14 | 99.04 | 98.92 | 98.62 | 97.08 | 84.66 | 22.21 |
| | MAP | MAP | 99.01 | 99.01 | 98.99 | 98.96 | 98.93 | 98.85 | 98.57 | 97.69 | 88.82 | 15.42 |
| Random | Diag | parameter-wise | 55.88 | 25.95 | 11.15 | 10.88 | 11.13 | 10.56 | 11.88 | 11.35 | 9.95 | 9.81 |
| | | layerwise | 78.86 | 17.47 | 22.35 | 14.23 | 12.06 | 11.80 | 10.18 | 12.82 | 9.35 | 9.80 |
| | | scalar | 88.75 | 60.17 | 25.35 | 15.98 | 14.26 | 11.91 | 8.63 | 9.74 | 9.05 | 9.80 |
| | KFAC | layerwise | 89.90 | 18.70 | 20.36 | 14.13 | 14.56 | 12.61 | 9.72 | 12.00 | 8.93 | 9.80 |
| | | scalar | 90.46 | 34.14 | 19.98 | 12.72 | 8.49 | 11.40 | 10.08 | 10.54 | 9.74 | 9.80 |
| | MAP | MAP | 79.03 | 43.25 | 22.86 | 9.68 | 10.34 | 9.96 | 11.50 | 9.50 | 10.85 | 9.80 |

Table E.1: Comparison of pruning accuracies of SpaM training with different pruning criteria, Hessian approximations, and priors for post-hoc pruning LeNet on MNIST.

| Method | Approximation | Sparsity (%) Prior | 20 | 40 | 60 | 70 | 75 | 80 | 85 | 90 | 95 | 99 |
|---|---|---|---|---|---|---|---|---|---|---|---|---|
| **GraSP** | Diag | parameter-wise | **98.51** | **98.51** | **98.51** | **98.51** | **98.51** | 98.51 | **98.52** | 98.51 | **98.52** | **98.50** |
| | | layerwise | 97.71 | 97.71 | 97.71 | 97.71 | 97.71 | 97.71 | 97.71 | 97.71 | 97.71 | 90.35 |
| | | scalar | 97.91 | 97.91 | 97.91 | 97.91 | 97.91 | 97.91 | 97.91 | 97.91 | 97.91 | 92.54 |
| | KFAC | parameter-wise | 98.10 | 98.10 | 98.10 | 98.10 | 98.10 | 98.10 | 98.10 | 98.10 | 98.11 | 95.45 |
| | | layerwise | 98.16 | 98.16 | 98.16 | 98.16 | 98.16 | 98.16 | 98.16 | 98.15 | 97.92 | 57.14 |
| | | scalar | 98.29 | 98.29 | 98.29 | 98.29 | 98.29 | 98.29 | 98.29 | 98.29 | 98.21 | 64.88 |
| | MAP | MAP | 98.41 | 98.41 | 98.38 | 98.38 | 98.33 | 98.35 | 98.17 | 97.38 | 89.43 | 38.89 |
| **SNIP** | Diag | parameter-wise | **98.51** | **98.51** | **98.51** | **98.51** | **98.51** | **98.52** | 98.49 | 97.70 | 83.76 | 20.00 |
| | | layerwise | 97.71 | 97.71 | 97.71 | 97.71 | 97.71 | 97.71 | 97.71 | 97.71 | 97.71 | 12.41 |
| | | scalar | 97.91 | 97.91 | 97.91 | 97.91 | 97.91 | 97.91 | 97.91 | 97.91 | 97.91 | 75.84 |
| | KFAC | parameter-wise | 98.10 | 98.10 | 98.10 | 98.10 | 98.10 | 98.10 | 98.10 | 98.10 | 97.84 | 32.84 |
| | | layerwise | 98.16 | 98.16 | 98.16 | 98.16 | 98.15 | 98.15 | 98.14 | 98.10 | 92.04 | 28.16 |
| | | scalar | 98.29 | 98.29 | 98.29 | 98.29 | 98.29 | 98.29 | 98.29 | 98.29 | 97.89 | 54.69 |
| | MAP | MAP | 98.38 | 98.36 | 98.34 | 98.22 | 98.02 | 97.44 | 96.04 | 92.22 | 81.42 | 35.36 |
| **OPD** | Diag | parameter-wise | **98.51** | **98.51** | **98.51** | **98.51** | **98.51** | 98.51 | **98.52** | **98.52** | 98.51 | **98.50** |
| | | layerwise | 97.71 | 97.71 | 97.71 | 97.71 | 97.71 | 97.71 | 97.71 | 97.71 | 97.71 | 96.06 |
| | | scalar | 97.91 | 97.91 | 97.91 | 97.91 | 97.91 | 97.91 | 97.91 | 97.91 | 97.91 | 96.47 |
| | KFAC | parameter-wise | 98.10 | 98.10 | 98.10 | 98.10 | 98.10 | 98.10 | 98.10 | 98.10 | 98.10 | 96.97 |
| | | layerwise | 98.16 | 98.16 | 98.16 | 98.16 | 98.16 | 98.16 | 98.16 | 98.15 | 97.84 | 86.81 |
| | | scalar | 98.29 | 98.29 | 98.29 | 98.29 | 98.29 | 98.29 | 98.29 | 98.29 | 98.23 | 84.91 |
| | MAP | MAP | 98.38 | 98.39 | 98.38 | 98.35 | 98.34 | 98.32 | 98.20 | 97.72 | 94.26 | 57.66 |
| **Magnitude** | Diag | parameter-wise | **98.51** | **98.51** | **98.51** | **98.51** | **98.51** | **98.52** | **98.52** | 98.51 | **98.52** | 98.48 |
| | | layerwise | 97.71 | 97.71 | 97.71 | 97.71 | 97.71 | 97.71 | 97.71 | 97.71 | 97.70 | 76.89 |
| | | scalar | 97.91 | 97.91 | 97.91 | 97.91 | 97.91 | 97.91 | 97.91 | 97.91 | 97.91 | 96.24 |
| | KFAC | parameter-wise | 98.10 | 98.10 | 98.10 | 98.10 | 98.10 | 98.10 | 98.10 | 98.10 | 98.10 | 94.22 |
| | | layerwise | 98.16 | 98.16 | 98.16 | 98.16 | 98.16 | 98.16 | 98.16 | 98.14 | 97.89 | 36.31 |
| | | scalar | 98.29 | 98.29 | 98.29 | 98.29 | 98.29 | 98.29 | 98.29 | 98.29 | 98.24 | 82.57 |
| | MAP | MAP | 98.38 | 98.39 | 98.38 | 98.36 | 98.34 | 98.30 | 98.16 | 97.77 | 93.56 | 53.16 |
| **Random** | Diag | parameter-wise | 90.09 | 62.13 | 39.76 | 31.51 | 19.52 | 22.50 | 18.93 | 14.84 | 15.57 | 11.81 |
| | | layerwise | 83.18 | 53.42 | 32.12 | 28.44 | 22.51 | 22.82 | 18.88 | 17.00 | 12.95 | 11.02 |
| | | scalar | 85.66 | 57.33 | 37.74 | 25.43 | 23.45 | 20.28 | 22.31 | 16.64 | 14.89 | 9.85 |
| | KFAC | parameter-wise | 85.95 | 58.41 | 39.27 | 27.96 | 24.64 | 21.94 | 18.72 | 17.04 | 13.66 | 9.37 |
| | | layerwise | 87.00 | 58.40 | 41.93 | 33.39 | 30.62 | 27.91 | 20.77 | 21.02 | 14.07 | 10.49 |
| | | scalar | 90.16 | 64.54 | 40.36 | 34.11 | 30.63 | 28.92 | 17.88 | 18.35 | 13.82 | 11.47 |
| | MAP | MAP | 97.26 | 87.54 | 65.94 | 52.32 | 47.22 | 44.12 | 40.72 | 22.46 | 16.59 | 8.87 |

Table E.2: Comparison of pruning accuracies of SpaM training with different pruning criteria, Hessian approximations, and priors for post-hoc pruning MLP-Mixer (2 blocks) on MNIST.

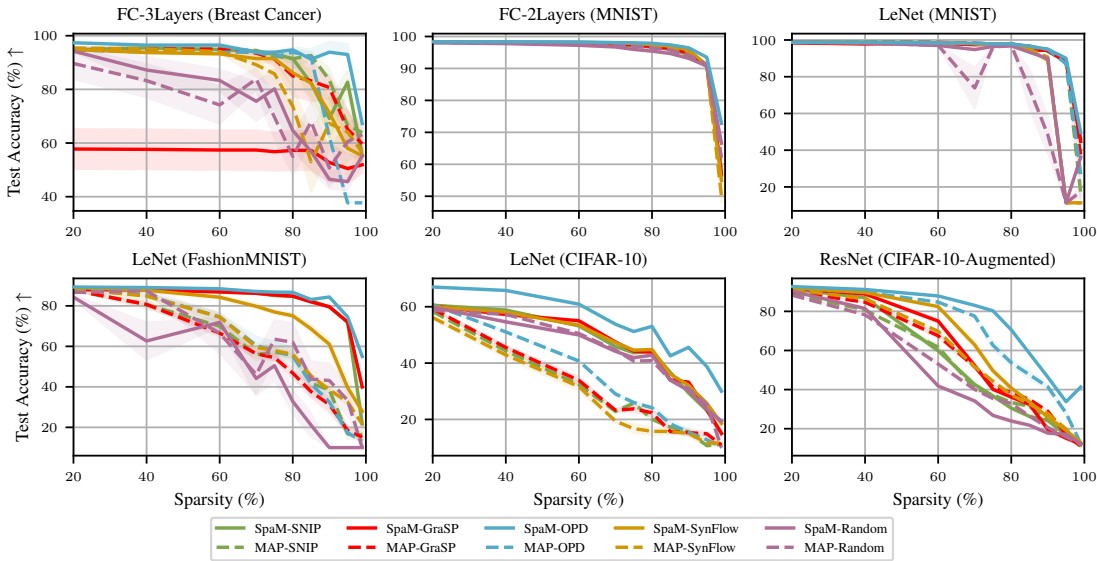

Figure E.3: Similarly to unstructured pruning, we also see in this experiment on structured pruning that SpaM (using a unit-wise prior) improves performance over MAP and that OPD mostly outperforms other pruning criteria, especially at higher sparsity levels.

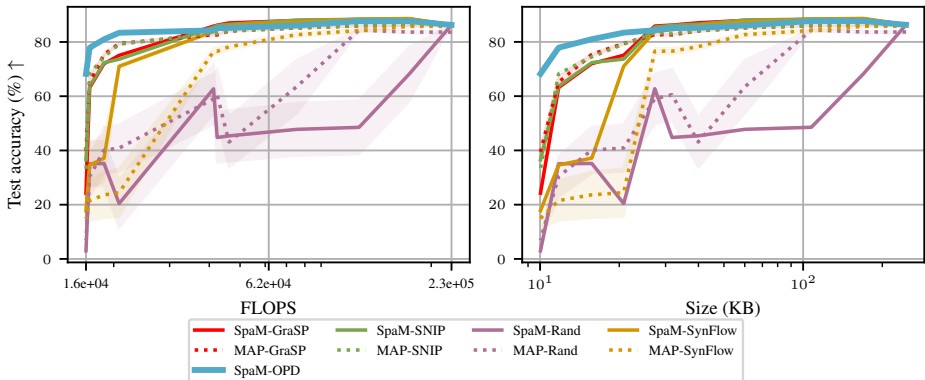

Figure E.4: Structured pruning with LeNet on FashionMNIST, using unit-wise priors. We see that our SpaM-OPD dominates the Pareto frontier in terms of predictive performance as a function of computational time and memory cost.

## E.6. Unit-wise and Parameter-wise KFAC for GraSP

As shown in Section 3, networks trained using SpaM and parameter-wise priors were able to maintain a high accuracy at challenging sparsity levels up to 99%. Moreover, parameter-wise KFAC and unit-wise priors showed high performance for the OPD pruning approach. We show in Figure E.7 that the combination of SpaM and these priors leverage the performance of methods like GraSP.

Table E.3: NLL of unstructured pruned ResNets on CIFAR-10. The best training method for each pruning criterion is highlighted in green, showing that SpaM improves performance over MAP for most criteria. The best performances (lowest NLL) overall at each sparsity level are shown in **bold**, showing that our OPD criterion outperforms the others at most sparsity levels.

| Criterion | Training | Sparsity (%) | | | | | | |
|---|---|---|---|---|---|---|---|---|
| | | 70 | 75 | 80 | 85 | 90 | 95 | 99 |
| OPD | MAP | 0.53 ± 0.0013 | 0.52 ± 0.0011 | 0.54 ± 0.0011 | 0.69 ± 0.0036 | 1.31 ± 0.0086 | 2.08 ± 0.0190 | 2.62 ± 0.0106 |
| | SpaM | **0.36 ± 0.0016** | **0.36 ± 0.0016** | **0.37 ± 0.0014** | **0.38 ± 0.0022** | **0.44 ± 0.0056** | **0.80 ± 0.0270** | 3.43 ± 0.0179 |
| GraSP | MAP | 0.51 ± 0.0008 | 0.54 ± 0.0032 | 0.66 ± 0.0046 | 1.11 ± 0.0195 | 1.73 ± 0.0193 | 2.35 ± 0.0276 | 2.69 ± 0.0088 |
| | SpaM | 0.37 ± 0.0007 | 0.38 ± 0.0006 | 0.40 ± 0.0015 | 0.42 ± 0.0032 | 0.51 ± 0.0093 | 0.97 ± 0.0317 | 3.71 ± 0.0709 |
| Magnitude | MAP | 0.54 ± 0.0014 | 0.53 ± 0.0011 | 0.55 ± 0.0015 | 0.73 ± 0.0034 | 1.54 ± 0.0098 | 2.65 ± 0.0239 | 2.70 ± 0.0113 |
| | SpaM | 0.37 ± 0.0011 | 0.37 ± 0.0012 | 0.38 ± 0.0016 | 0.41 ± 0.0028 | 0.49 ± 0.0072 | 0.92 ± 0.0320 | 3.63 ± 0.0418 |
| Random | MAP | 2.79 ± 0.0438 | 2.63 ± 0.0089 | 2.42 ± 0.0146 | 2.36 ± 0.0042 | 2.33 ± 0.0037 | 2.34 ± 0.0043 | **2.30 ± 0.0000** |
| | SpaM | 2.60 ± 0.0292 | 3.22 ± 0.0701 | 2.60 ± 0.0230 | 2.70 ± 0.0447 | 2.38 ± 0.0044 | 2.31 ± 0.0009 | 2.31 ± 0.0003 |
| SNIP | MAP | 1.54 ± 0.0595 | 1.45 ± 0.0235 | 2.18 ± 0.0331 | 2.51 ± 0.0350 | 2.90 ± 0.0450 | 4.44 ± 0.0864 | 3.28 ± 0.0203 |
| | SpaM | 0.84 ± 0.0320 | 1.27 ± 0.0474 | 2.01 ± 0.0814 | 2.93 ± 0.1060 | 3.72 ± 0.1244 | 3.97 ± 0.0907 | 3.26 ± 0.0841 |

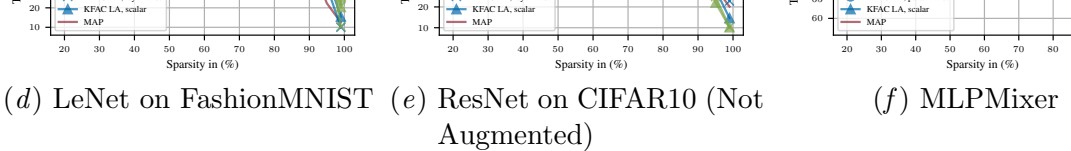

(a) FC on Cancer  (b) FC on MNIST  (c) LeNet on MNIST

(d) LeNet on FashionMNIST  (e) ResNet on CIFAR10 (Not Augmented)  (f) MLPMixer

Figure E.5: Effect of different priors and Hessian approximations on the sparsification performance with SpaM-OPD. The diagonal approximation with parameter-wise priors is a strong choice, especially at higher sparsities, while the KFAC approximation with layerwise prior yields slightly better performances at lower sparsities.

### E.7. Online pruning.

### E.8. One Shot Efficiency

As seen in Figure E.9, our proposed post-hoc pruning criterion, OPD, consistently demonstrates stable performance across diverse model architectures and datasets, achieving significant sparsity levels without the need for fine-tuning. It seamlessly operates either post-training or with pre-trained models, providing a highly flexible and versatile solution.

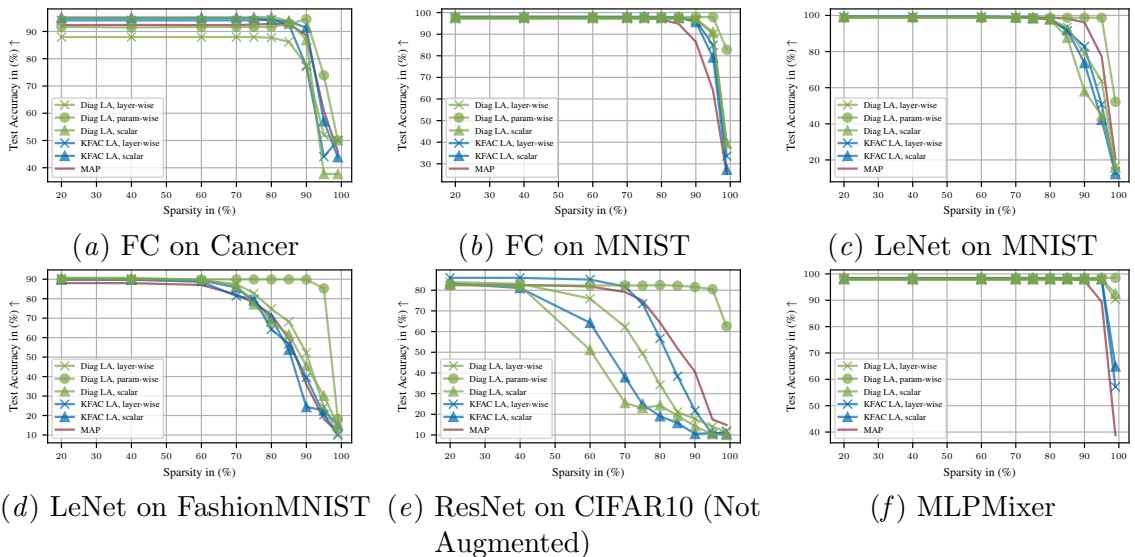

Figure E.6: Priors and Hessian approximations for GraSP pruning. We see that the effects are qualitatively similar to pruning with OPD.

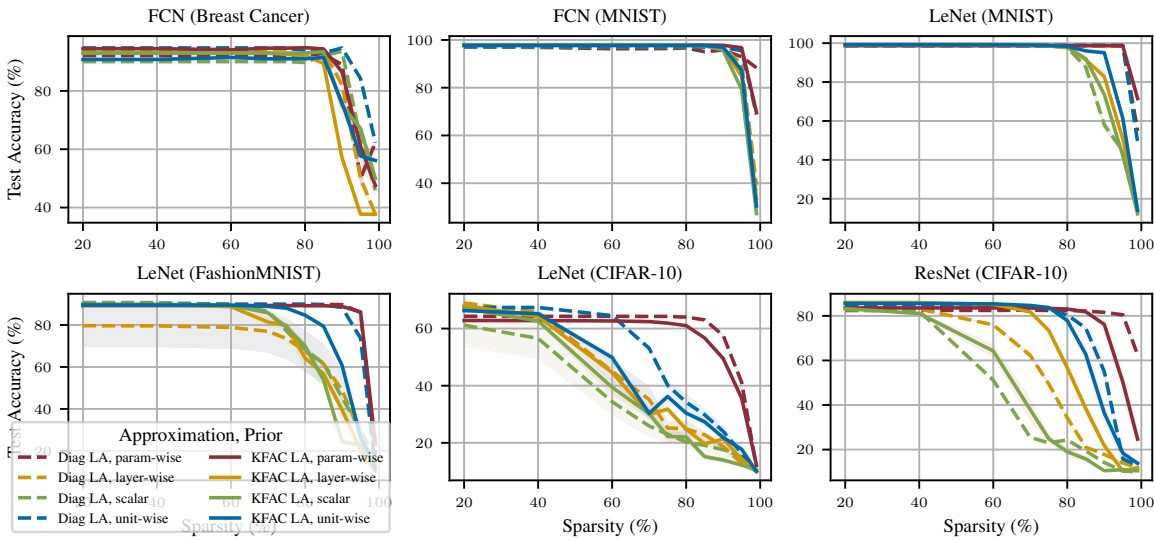

Figure E.7: Priors and Hessian approximations for GraSP pruning with SpaM. We see that the effects are qualitatively similar to pruning with OPD.

## E.9. Modern Architectures

### E.9.1. WIDE RESNET

In Figure E.10, we demonstrate how SpaM enhances the sparsity performance of Wide ResNet models. This is specifically illustrated in the case of OPD, GraSP, and Magnitude,

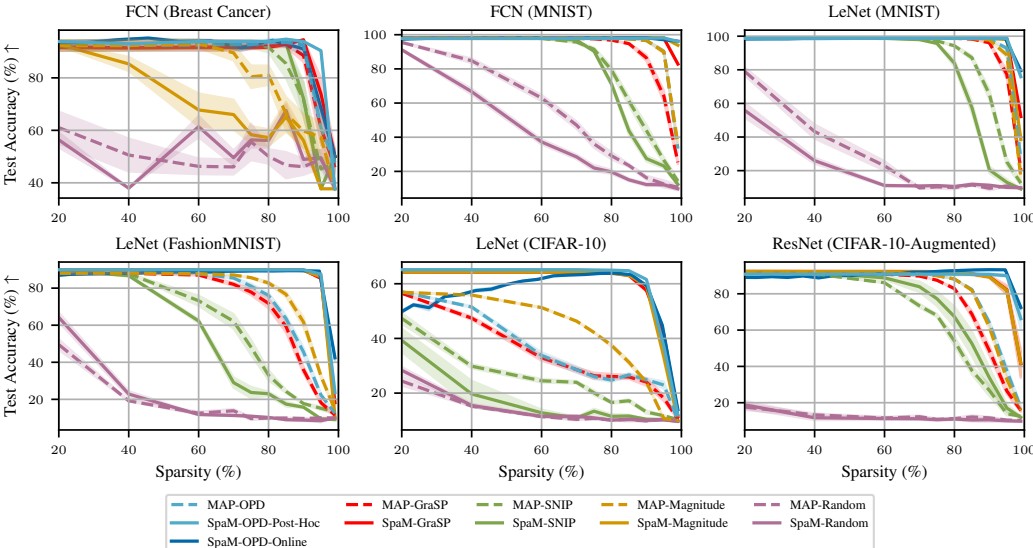

Figure E.8: Predictive performance as a function of sparsity level in unstructured pruning. We include our online pruning approach that progressively prunes **a** model during the training compared to the other curves demonstrating the performance of 10 pruned models based on **a converged** baseline. our online pruning approach is often competitive with post-hoc pruning.

all while maintaining a low Brier score, ECE, and NLL up to 95% sparsity. On a larger dataset like CIFAR100, We observe the same trend as shown in Figure E.11

### E.9.2. VISION TRANSFORMER

Figure E.12 demonstrates the impact of SpaM on Unstructured Pruning for a Vision Transformer (ViT) trained on MNIST. These results align with the findings presented in Section 3 with SpaM diagonal LA and parameter-wise priors leveraging the sparsifiability of models using OPD, Magnitude and GraSP, maintaining a test accuracy of 97% for OPD and Magnitude at 95% sparsity compared to an accuracy of lower than 20% under MAP for the same methods. This serves as a proof-of-concept for vision transformers but efficacy has to be verified at a larger scale where such models perform best.

### E.9.3. GPT-2

We demonstrate the efficacy of OPD on a pre-trained GPT-2 model (124M parameters) fine-tuned for sentiment analysis on the IMDB dataset. To manage computational resources, we limit both the Laplace approximation and SpaM to two steps. Despite this constraint, OPD maintains high predictive performance even at 60% sparsity, as shown in Figure E.13. This suggests that extending SpaM optimization with more epochs and a more refined posterior could further enhance performance.

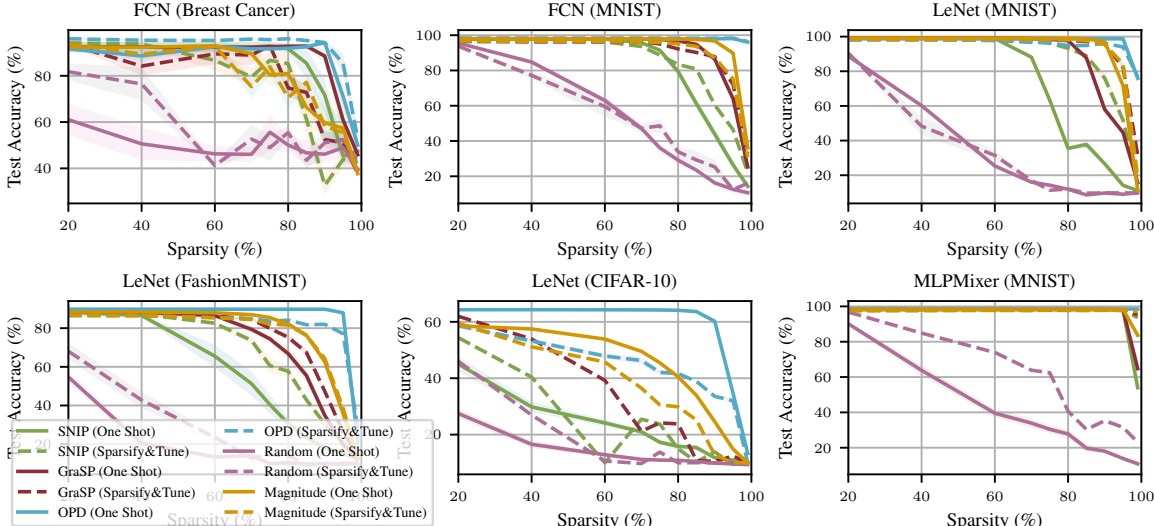

Figure E.9: SpaM post-hoc pruning efficiency with optional fine-tuning after the pruning. Unlike other pruning criteria, OPD does not require additional tuning to achieve optimal performance across different architectures and often still outperforms the other fine-tuned methods.

## E.10. Visualization of Pruning Process

To visualize the model's structural evolution during pruning, we present a series of filter bank visualizations that capture the key stages of transformation, from the initial dense architecture to the final compact form (Figures E.14, E.15, E.16, E.17).

## E.11. Network Compression

In Figures E.18 and E.19, we demonstrate the efficiency gains achieved by our SpaM-OPD approach. For the fully connected network on the Cancer dataset, it achieves a remarkable reduction of over 20 times in disk size and 24 times in FLOPs while simultaneously maintaining baseline test accuracy. Additionally, it boasts a Brier score of 0.15 and a negative log marginal likelihood (Neg Log MargLik) lower than the original model. These results highlight the effectiveness of SpaM-OPD in achieving significant model compression without compromising performance on key metrics.

## E.12. Computational Cost

Instead of using GGN approximation, which scales linearly with the number of classes, we can also use the EF. Using EF instead of GGN for SpaM does not add computational overhead compared to MAP, as EF costs roughly as much as gradient computation. The pruning results are not affected by the choice of GGN or EF. The runtime of MAP and SPAM was identical (roughly 1h and 20 minutes on A100s) for WRN-16 on Cifar100 using SpaM (EF) diagonal LA with parameter-wise prior (our recommended settings for pruning)

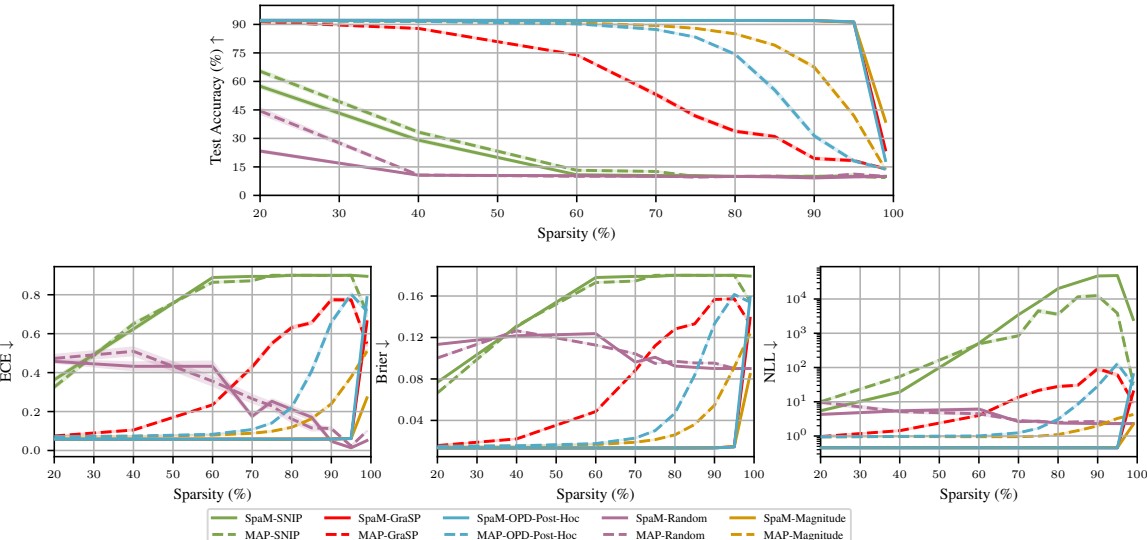

Figure E.10: SpaM post-hoc efficiency for Wide ResNet 16 on CIFAR10. Leveraging OPD, GraSP, and Magnitude performance under SpaM in comparison to MAP show superior test accuracy at increased sparsity levels coupled with a low ECE, Brier Score, and NLL.

in comparison to MAP training. In prior works (Immer et al., 2021b), it was found that GGN gives a better posterior predictive approximation, but we do not use it in this work. We find that EF works similarly well for pruning at a much lower cost.

## Appendix F. Technical Details

### F.1. Resizing and Compression

Post structured pruning, the model may undergo fine-tuning to regain performance. In this process, pruned structures are completely removed from the architecture rather than merely being zeroed out. This leads to a network with fewer filters in convolutional layers and a reduced number of neurons in fully connected layers, resulting in a leaner and more efficient model.

The process of compaction involves transferring the weights from the pruned model to a newly created, smaller architecture that is aligned with the dimensions of the retained active structures. This results in a denser, storage- and computation-optimized model.

Algorithm 1 summarizes this entire process of structured pruning and model compaction.

This approach transitions the model from a pruned state to a compact and optimized architecture. The final compressed model $M_{\text{compact}}$ not only retains essential predictive capabilities but is also further tuned for performance. The newly configured $M_{\text{compact}}$ is saved with updated parameters, ensuring efficient inference and ease of deployment, especially on resource-constrained edge devices.

The reduced memory footprint and FLOPS of $M_{\text{compact}}$ are particularly beneficial for deployment on edge devices with limited computational resources. When models exceed

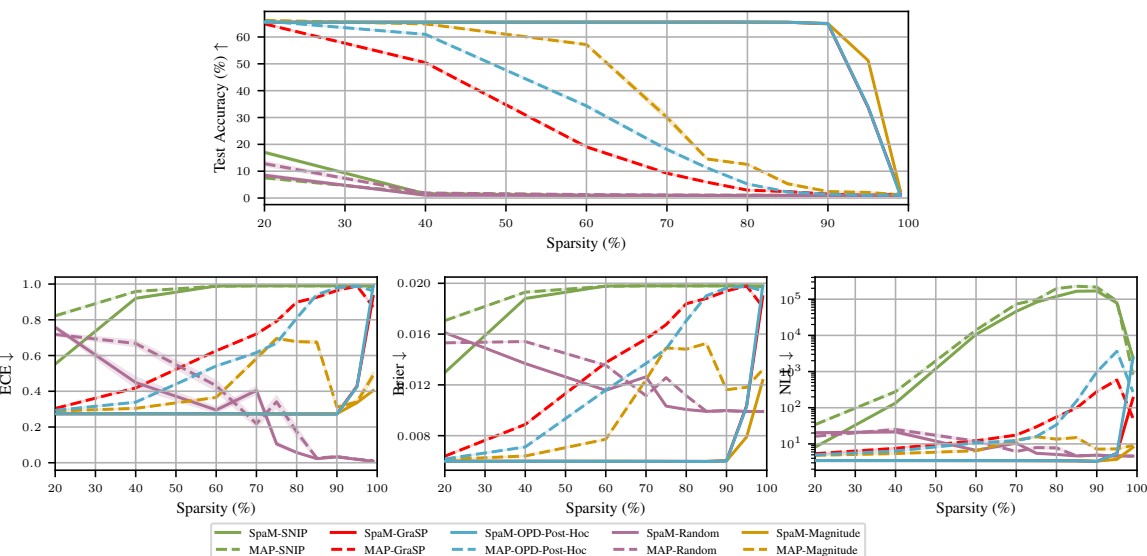

Figure E.11: SpaM post-hoc efficiency for Wide ResNet 16 on CIFAR100. Leveraging OPD, GraSP, and Magnitude performance under SpaM in comparison to MAP show superior test accuracy at increased sparsity levels coupled with a low ECE, Brier Score, and NLL.

the hardware limits, aggressive compression techniques like quantization may be required, which can compromise performance. Our method aims to significantly reduce the memory size of the model while minimizing performance trade-offs. The effectiveness of our approach in achieving this balance is explored in Section 3.

### F.2. Pseudocodes

Algorithm 1 outlines our structured pruning procedure, highlighting how we efficiently achieve a simpler model by transferring weights to a smaller one.

## Appendix G. Experimental Setup

### G.1. Datasets

*Breast Cancer Wisconsin (Diagnostic) (UCI)*: This dataset, derived from digitized images of fine needle aspirates of breast masses, includes features describing characteristics of cell nuclei in the images. It is a classic binary classification dataset used extensively in breast cancer research (Dua and Graff, 2019).

*MNIST*: A foundational benchmark dataset in machine learning, MNIST consists of 60,000 training and 10,000 test images of handwritten digits (0 to 9) in 28x28 pixel grayscale format (LeCun et al., 1998).

*FashionMNIST*: A drop-in replacement for MNIST, Fashion-MNIST offers a greater challenge with its 60,000 training and 10,000 test images in grayscale (28x28 pixels). Each image represents one of ten clothing categories (Xiao et al., 2017).

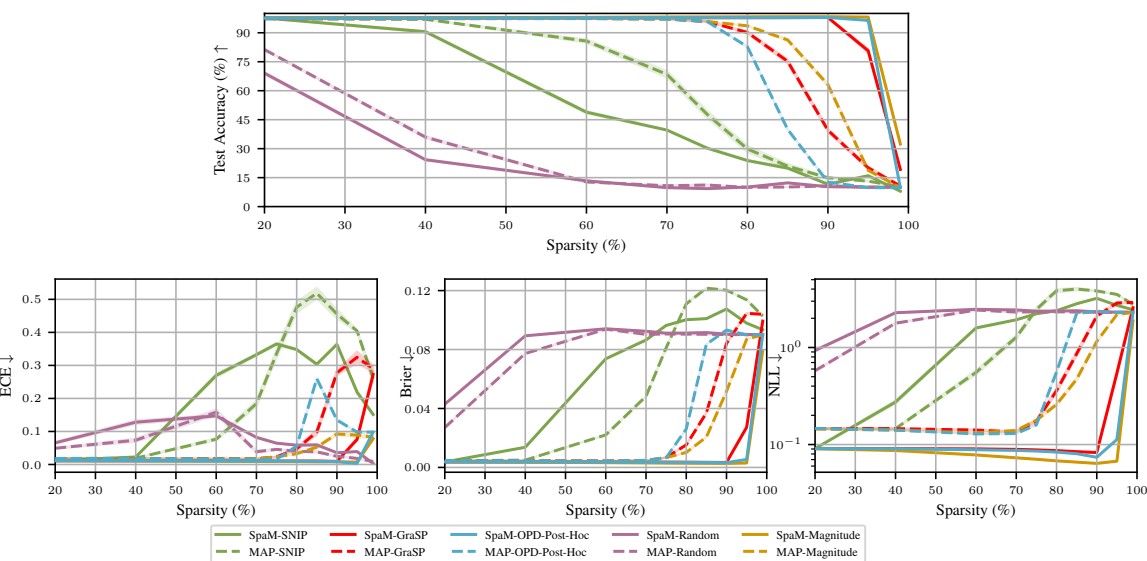

Figure E.12: SpaM post-hoc efficiency for ViT on MNIST. Leveraging OPD, GraSP, and Magnitude performance under SpaM in comparison to MAP show superior test accuracy at increased sparsity levels coupled with a low ECE, Brier Score, and NLL.

---

**Algorithm 1** Structured OPD pruning

---

**Require:** Trained Model $M$, Target Sparsity Threshold $T$
**Ensure:** Compacted Model $M_{\text{compact}}$, Count of Pruned Units $N_{\text{pruned}}$
 1: **for** each layer $l$ in $M$ **do**
 2:  **if** $l$ is not the output layer **then**
 3:   **for** each structure $s$ in layer $l$ **do**
 4:    Calculate $A_s = \sum_{i \in S} P_{ii} \cdot \theta_i^2$
 5:   **end for**
 6:   Sort structures in $l$ by $A_s$ in ascending order
 7:   Determine the number of structures to prune based on $T$
 8:   Prune determined number of structures with the lowest $A_s$ values
 9:  **end if**
10: **end for**
11: Update $N_{\text{pruned}}$ with the count of pruned structures
12: Fine-tune the pruned model $M$
13: Initialize $M_{\text{compact}}$ with dimensions aligned to the unpruned structures of $M$
14: Transfer weights from unpruned structures of $M$ to $M_{\text{compact}}$
15: Save $M_{\text{compact}}$ with updated parameters

---

*CIFAR-10*: This dataset contains 60,000 color images (32x32 pixels) divided equally among 10 classes (e.g., airplane, bird, cat) (Krizhevsky, 2009). For our ResNet experiments, we augment CIFAR-10 with random flipping and cropping.

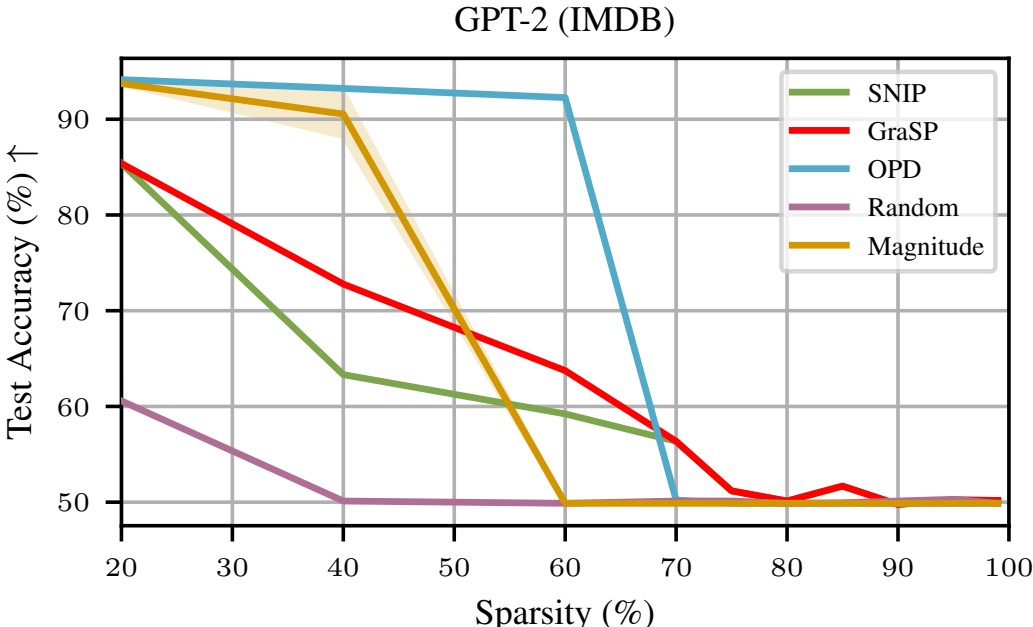

Figure E.13: GPT-2 (124M) on IMDB. We tune GPT-2 for sentiment analysis on IMDB datasets. Our results show that OPD maintains significantly higher accuracy than other methods, which degrade towards random classifier performance (50%) at 60% sparsities.

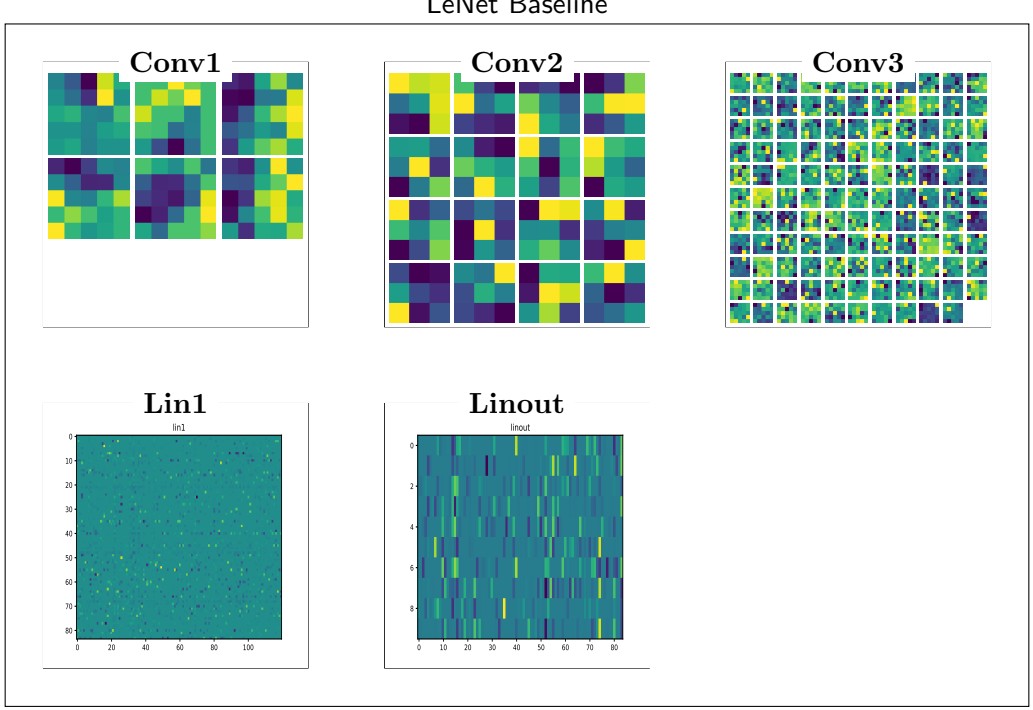

Figure E.14: Visualization of model weights for unpruned LeNet on FashionMNIST.

*CIFAR-100*: A more fine-grained version of CIFAR-10, this dataset includes 60,000 color images (32x32 pixels) across 100 classes, with 600 images per class (Krizhevsky, 2009). We apply random flipping and cropping for augmentation.

*IMDB Movie Review*: This dataset is a collection of 50,000 movie reviews, balanced between positive and negative sentiments. It is commonly used for binary sentiment classification tasks (Maas et al., 2011).

## G.2. Models

*FCN for MNIST (784, 256, 10)*: This Fully Connected Network (FCN) is specifically designed for the MNIST dataset. It comprises an input layer with 784 nodes, a hidden layer with 256 nodes, and an output layer with 10 nodes, making it a 2-layer FC network. Its architecture is optimized to handle the simplicity and characteristics of handwritten digit images.

*FCN for CANCER (30, 100, 2)*: Customized for the CANCER dataset, this FCN includes an input layer of 30 nodes, two hidden layers, each containing 100 nodes, and a final output layer of 2 nodes. The 3-layer structure of this network is instrumental in distinguishing between benign and malignant tumors based on cellular features.

*LeNet*: As a foundational Convolutional Neural Network (CNN), LeNet has shown exceptional performance in digit and image recognition tasks. We have applied LeNet to the MNIST, Fashion MNIST, and CIFAR-10 datasets, leveraging its capability to handle varying complexities of image data (LeCun et al., 1998). LeNet on CIFAR-10 is not a very

Figure E.15: Visualization of model weights for 95 % pruned LeNet on FashionMNIST. Black refers to pruned weights or a part of a filter. An entire black square refers to an entire filter being pruned. A black row or column represents a pruned neuron.

Figure E.16: Visualization of model weights for 95 % pruned LeNet on FashionMNIST after the zeroing stage. Black refers to pruned weights or a part of a filter. An entire black square refers to an entire filter being pruned. A black row or column represents a pruned neuron. (Model size = 241 KB).

Figure E.17: Visualization of model weights for 95 % pruned LeNet on FashionMNIST after the compression stage. Black refers to pruned weights or a part of a filter. An entire black square refers to an entire filter being pruned. A black row or column represents a pruned neuron. (Model size = 12 KB).

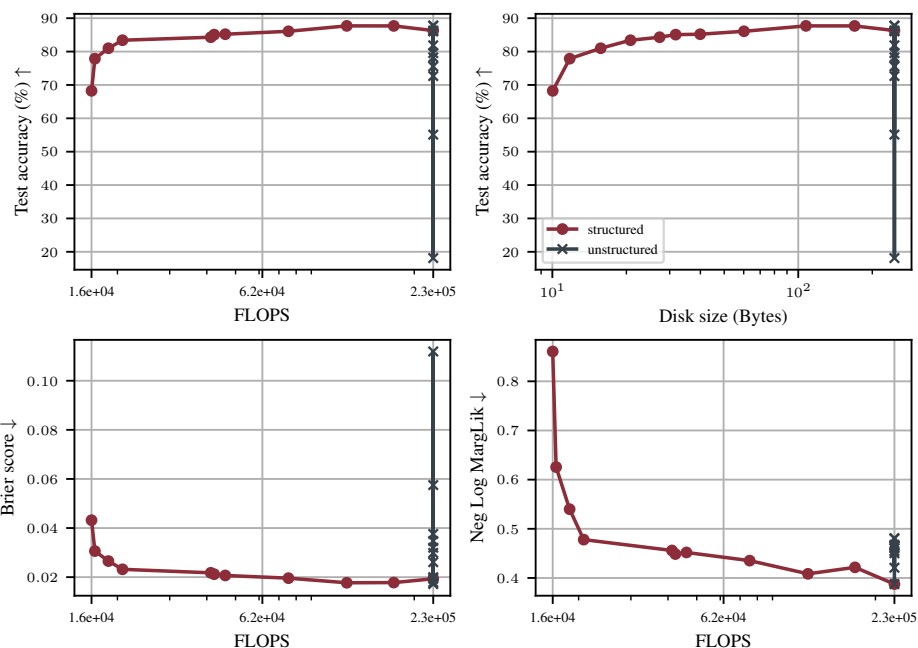

Figure E.18: Structured and unstructured pruning of LeNet on FashionMNIST with SpaM-OPD. We see that through structured sparsification, we are able to obtain models that are still performant at a significantly reduced computational and memory cost, while unstructured pruning does not directly translate into computational benefits.

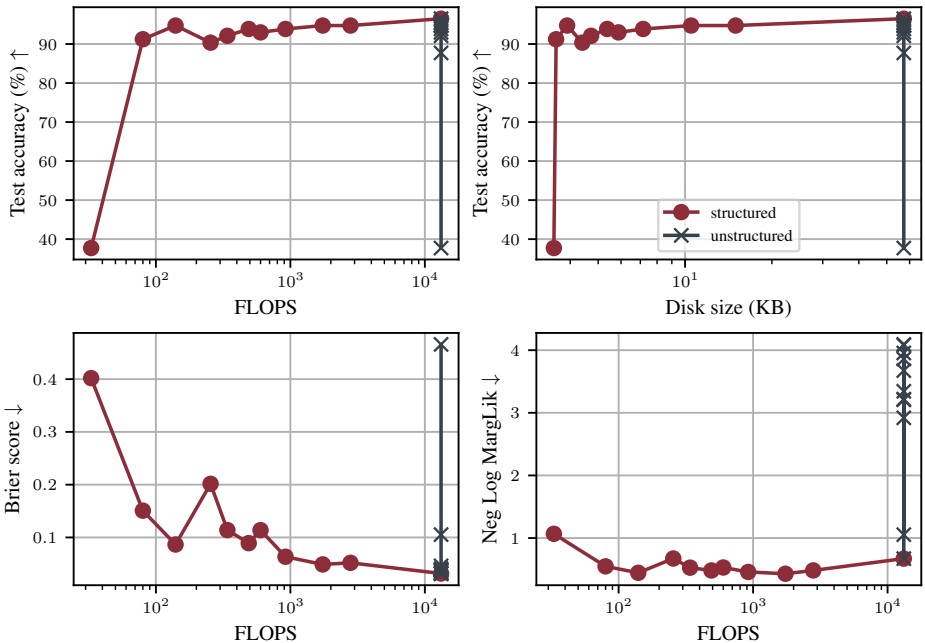

Figure E.19: Structured and unstructured pruning of FC on Cancer with SpaM-OPD. We see that through structured sparsification, we are able to obtain models that are still performant at a significantly reduced computational and memory cost. At the same time, unstructured pruning does not directly translate into computational benefits.

common benchmark for pruning; here, it is used to demonstrate how SpaM, and specifically SpaM-OPD, is able to prune at high percentages without a performance loss up to 80% in a model that struggles with representing the data's complexity, showing that our work extends beyond over-parametrized networks for the task at hand.

*MLPMixer*: The MLPMixer serves as a streamlined alternative to more complex models like CNNs and transformers. It relies solely on Multi-Layer Perceptrons (MLPs) for integrating inputs across spatial and channel dimensions (Tolstikhin et al., 2021). We implement an MLPMixer with 2 blocks designed for MNIST.

*ResNet with inplanes 64 and depth 18 for CIFAR-10*: We modify the implementation of ResNet and incorporate fixup initialization and custom bias and scale parameters to align with the constraints of the *ASDL* backend (Osawa et al., 2023) used for the Laplace computations in this work, which does not support batch normalization.

*Wide ResNet*: decreases depth compared to ResNet and increases the width of residual networks (Zagoruyko and Komodakis, 2017) with a depth of 16 and a widening factor of 4 (WRN16-4). We use fixup blocks to be able to utilize *ASDL* backend (Osawa et al., 2023).

*Vision Transformer* (ViT) (Dosovitskiy et al., 2021): unlike CNNs, which extract local features through filters and pooling layers, ViT breaks down images into fixed-size patches, treating each as a "token" in a sequence (Dosovitskiy et al., 2021). This allows it to

leverage the Transformer architecture, initially designed for language processing, to analyze relationships between patches through *self-attention* mechanisms (Vaswani et al., 2017).

*DistilBERT*: DistilBERT (Sanh et al., 2020) is a smaller, faster, and cheaper version of BERT, achieved by leveraging knowledge distillation during the pre-training phase. This model retains 97% of BERT's language understanding capabilities while being 60% faster and 40% smaller. We use the pre-trained DistilBERT hosted in Hugging Face under (`distilbert-base-uncased`) (Sanh et al., 2020) and tune it for sentiment analysis to classify reviews in the IMDB dataset (Maas et al., 2011) , which involves predicting the sentiment (positive or negative) of user reviews based on their textual content.

*GPT-2* : a large-scale transformer-based language model developed by OpenAI , with impressive text generation capabilities. Trained on a vast corpus of internet text (Radford et al., 2019). In our study, we leverage the 124M parameter version of GPT-2, fine-tuning it on the IMDB dataset for sentiment analysis to assess its performance under different pruning conditions.

### G.3. Dependencies

For the computation of second-order information (e.g., Hessian, Fisher information) needed for the Laplace approximation, we utilize the ASDL Library (Osawa et al., 2023). We use the library in its version under `https://github.com/kazukiosawa/asdl/tree/011a942b2698b9ec33b0c8c47c96` The ASDL Library is distributed under the MIT License, which allows for reuse with a few restrictions that we respect in our work.

### G.4. Hyperparameters

**Marginal Likelihood**

- Hessian Approximation: The choice between GGN and EF. GGN was initially employed for fully connected networks, LeNets, and ResNets. However, for complex architectures (WRNs, ViTs, DistilBERT), GGN's computational cost became prohibitive, exceeding MAP runtime by up to 20x and even more for casual modeling tasks. Switching to EF maintained pruning performance while closely matching MAP runtime, which is particularly beneficial as GGN scales linearly with the number of classes. We discuss further the cost in Appendix G.7.

- *n_epochs_burnin* Dictates the number of epochs after which marginal likelihood optimization starts. If set superior to the number of training epochs, marginal likelihood is skipped, and the training is equivalent to MAP.

- *marglik_frequency* Controls the frequency of marginal likelihood estimation. The default value of 1 signifies re-estimation after each epoch, while a value of 5 indicates approximation for every fifth epoch.

We use these parameters to manage the computational cost of our experiments, where for small models like LeNets, FC Networks, the *n_epochs_burnin* is set to zero and *marglik_frequency* to one reflecting estimating each epoch since the start of the training. In contrast, for complex networks like MLPMixer, ResNets, WideResNet, and ViT that we train from scratch, we start after 20 epochs and at an interval frequency of 5 epochs.

**Hyperparameter Table** Table G.1 presents the specific hyperparameters employed for each dataset-architecture combination. We use † to denote the use of data augmentation in the training process. The symbols ⋆ and ⋄ represent the use of the Generalized Gauss-Newton (GGN) and Empirical Fisher (EF) approximations for the Hessian, respectively. We use *cosine decay* scheduler towards a fixed *minimum learning rate* of 1e-6 across all experiments. The symbols $\mathcal{D}_1$, $\mathcal{D}_2$, etc., represent the following datasets:

* $\mathcal{D}_1$: Breast Cancer Wisconsin (Diagnostic) * $\mathcal{D}_2$: MNIST * $\mathcal{D}_3$: FashionMNIST * $\mathcal{D}_4$: CIFAR-10 * $\mathcal{D}_5$: CIFAR-100 * $\mathcal{D}_6$: IMDB Movie Review

All models are trained from scratch, denoted by the symbol ▲, except for DistilBERT and GPT-2, which are fine-tuned from pre-trained weights and are indicated by ▼.

| Dataset (Arch.) | Marglik Freq. | Batch Size | Learning Rate | Optimizer | Temp. | Burn-in / Epochs |
|---|---|---|---|---|---|---|
| $\mathcal{D}_1^{▲}$ (FCN) | 1 ⋆ | 64 | 0.001 | Adam | 1.0 | 0 / 50 |
| $\mathcal{D}_2^{▲}$ (FCN) | 1 ⋆ | 64 | 0.001 | Adam | 1.0 | 0 / 100 |
| $\mathcal{D}_2^{▲}$ (LeNet) | 1 ⋆ | 128 | 0.001 | SGD | 1.0 | 0 / 100 |
| $\mathcal{D}_3^{▲}$ (LeNet) | 1 ⋆ | 128 | 0.001 | SGD | 1.0 | 0 / 100 |
| $\mathcal{D}_3^{▲}$ (MLPMixer) | 1 ⋆ | 128 | 0.001 | Adam | 1.0 | 0 / 100 |
| $\mathcal{D}_4^{†}$ (ResNet) | 5 ⋆ | 128 | 0.1 | SGD | 5 | 20 / 100 |
| $\mathcal{D}_5^{†}$ (WRN) | 5 ⋄ | 128 | 0.1 | SGD | 5 | 20 / 200 |
| $\mathcal{D}_2^{▲}$ (ViT) | 5 ⋄ | 128 | 0.001 | Adam | 1.0 | 20 / 100 |
| $\mathcal{D}_6^{▼}$ (DistilBERT) | 5 ⋄ | 32 | 2e-5 | AdamW | 1.0 | 5 / 20 |
| $\mathcal{D}_6^{▼}$ (GPT-2) | 5 ⋄ | 8 | 2e-5 | Adam | 1.0 | 5 / 10 |

Table G.1: Hyperparameters used in the experiments.

**Computational resources** Our experiments are run on GPUs. We run our experiments in a single GPU configuration on available variation between 1080 Tis, V100s, and A100s, with the majority being run on A100s with 40GB memory as we run the experiments intensively one after the other for different architecture on the same allocated GPU and in order to provide enough GPU memory. For models such as FCs, LeNets, ResNets, and MLP-Mixer, a GPU with 12GB of memory ( 1080 Ti) proved sufficient to run our method for our recommended laplace and prior, which is diagonal with parameter-wise priors and reproduce the results. For the sentiment analysis task using GPT-2, we recommend using a 32 GB GPU for tuning to be able to utilize a high batch size and to use diagonal approximation to fit laplace on the data without running into memory shortage.

**Runtime Table**

Table G.2 presents the training and pruning runtimes on A100 for each dataset-architecture combination. Training times are given for both SpaM diagonal with parameter-wise prior and MAP, while pruning time is identical to both. Pruning runtimes refer to the time taken for OPD to compute and prune a model at 10 target sparsities. OPD and magnitude are very close in terms of runtime and the most efficient compared to SNIP, which is slightly slower due to it requiring an additional forward pass, and GraSP, which is significantly slower as it accumulates the gradient as shown in Figure G.1.

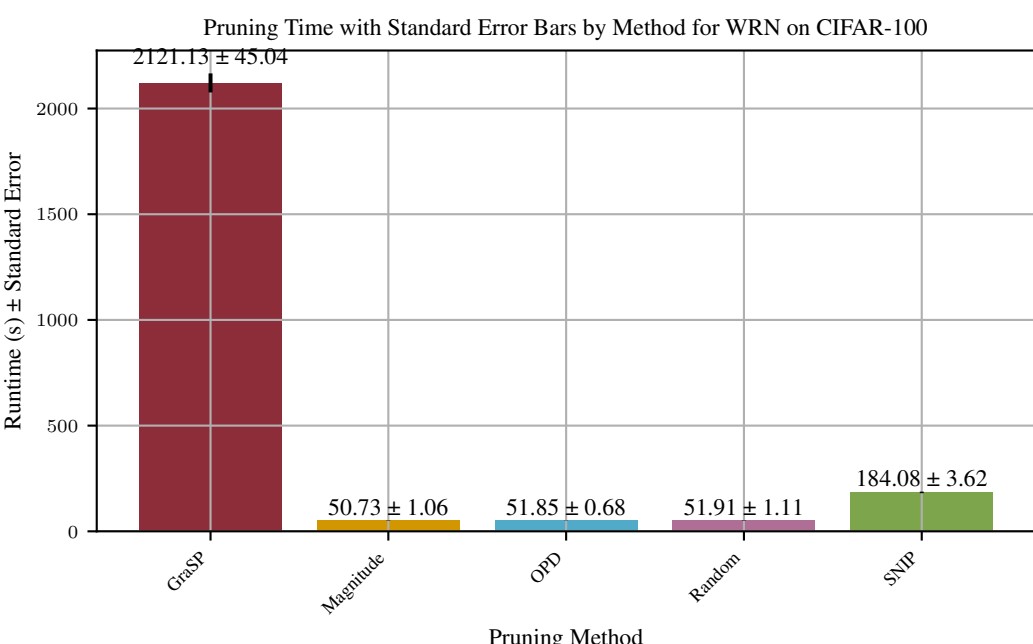

Figure G.1: Mean relative pruning time with standard error bars on WRN with CIFAR-100. OPD and Magnitude are the most efficient as they use pre-computed parameters, with SNIP being slightly slower due to requiring an additional forward pass, while GraSP is significantly slower as it needs to accumulate the gradient.

| Dataset (Arch.) | Train | | Prune |
|---|---|---|---|
| | SpaM | MAP | OPD |
| $\mathcal{D}_1^{\blacktriangle}$ (FCN) | 0:01 | 0:01 | 0:04 |
| $\mathcal{D}_2^{\blacktriangle}$ (FCN) | 0:15 | 0:5 | 0:23 |
| $\mathcal{D}_2^{\blacktriangle}$ (LeNet) | 0:16 | 0:10 | 0:15 |
| $\mathcal{D}_3^{\blacktriangle}$ (LeNet) | 0:16 | 0:10 | 0:21 |
| $\mathcal{D}_3^{\blacktriangle}$ (MLPMixer) | 0:05 | 0:07 | 0:10 |
| $\mathcal{D}_4^{\dagger}$ (ResNet) | 1:24 | 0:25 | 0:55 |
| $\mathcal{D}_5^{\dagger}$ (WRN) | 1:17 | 1:12 | 0:51 |
| $\mathcal{D}_2^{\blacktriangle}$ (ViT) | 0:26 | 0:15 | 0:24 |
| $\mathcal{D}_6^{\blacktriangledown}$ (DistilBERT) | 5:20 | 2:15 | 1:05 |
| $\mathcal{D}_6^{\blacktriangledown}$ (GPT-2) | 17:34 | 6:24 | 17:41 |

Table G.2: Runtimes for the experiments. Train: Training time (h:m), Prune: Pruning time (m:s).

## G.5. Pruning Criteria

- **SNIP**: Uses connection sensitivity, *how much a specific weight contributes to the output loss*, for effective pruning (Lee et al., 2018).

- **GraSP**: Employs gradient signal preservation. GraSP relates to the concept of Gradient Flow (GF), defined as:

$$GF = gL(\Theta)^T gL(\Theta) = ||gL(\Theta)||_2^2, \tag{G.1}$$

emphasizing the impact of pruning on the training dynamics (Wang et al., 2020). We replicate the GraSP implementation of Rachwan et al. (2022), where we consider the absolute value of the importance score initially proposed by Lubana and Dick (2021) given by:

$$I(\Theta_t) = |\Theta_t^T H_L(\Theta_t) g_L(\Theta_t)| \tag{G.2}$$

Note that while the importance score was initially used before training, we propose to use this importance score as a one-shot criterion after the training process and show how SpaM can leverage the performance of GraSP.

- **Structured-SynFLow**: We challenge the capabilities of SynFlow (Tanaka et al., 2020), a data-agnostic pruning approach that prevents layer collapse that happens at high sparsities where layers are no longer able to perform at the model's predictive power. This typically occurs when the pruning algorithm, intentionally or inadvertently, removes a significant portion of weights or filters from a specific layer, effectively collapsing its functionality (Tanaka et al., 2020). We push SynFlow to its limits through advanced structured pruning strategies, where we prune layers aggressively at the same target sparsity, which facilitates the compression process and resizing. By applying rigorous layer-specific filtering and neuron pruning, we aim to test the

robustness and effectiveness of SynFlow in extreme sparsity *structured* scenarios. This approach not only benchmarks SynFlow's performance under stringent conditions but also explores its potential to maintain network functionality and accuracy in highly sparse neural network architectures.

- **Magnitude Pruning**: Relies on the magnitude of weights for pruning, aiming to maintain model performance while reducing complexity (Han et al., 2015). After the success shown by Han et al. (2015), many methods adapted magnitude as a pruning criterion coupled with different scheduling (Bellec et al., 2018; Mostafa and Wang, 2019; Zhou et al., 2020).

- **Random Pruning**: Prune weights or structure randomly.

### G.6. Structured Sparsification Process

For structured sparsification, contrasting with the unstructured approach, the process necessitates reshaping the weight matrices to effectively reduce model complexity. The steps include:

1. One-shot structure masking based on aggregated importance scores.

2. Continue training for five epochs using the model from Step 1 for preliminary evaluation.

3. Implementing two software design approaches:

   - In-place layer replacement in the model with smaller ones fitting the non-masked regions.
   - Creating a new, flexible model initialized to match the dimensions of the non-masked areas, requiring repeated reading of the nonzero mask for state-dictionary and metadata alignment.

4. Transferring non-zero structures to smaller layers and tuning the model.

Post structure removal, we extend the training phase to adapt the model weights and re-evaluate, ensuring seamless functionality once transferred to smaller layers. Particularly after significant structural reduction, our primary objective shifts to maximizing performance in the downsized model. This fine-tuning spans 5 or 10 epochs depending on the complexity of the original model's structure, which was initially trained for either 50 or 100 epochs.

### G.7. Computational Cost

Instead of using the Generalized Gauss-Newton (GGN) approximation, which scales linearly with the number of classes, we can also use the Empirical Fisher (EF). For most architectures, using EF instead of GGN for SpaM does not add a very large computational overhead to MAP, as EF costs roughly as much as gradient computation. This is particularly beneficial as GGN scales linearly with the number of classes. The pruning results are not significantly affected by the choice of GGN or EF.

The runtime of MAP and SpaM was close (roughly 1h and 20 minutes on A100s) for WRN-16 on CIFAR100 using SpaM (EF) with diagonal LA and parameter-wise priors (our recommended settings for pruning). For language transformers, specifically DistilBERT and GPT-2, SpaM with EF does result in a longer training time compared to MAP. However, this increase is considerably less than when using GGN, where a single epoch can take longer than the entire SpaM training with EF.

In prior works (Immer et al., 2021b), it was found that GGN gives a better posterior predictive approximation, but we do not use it in this work. We find that EF works similarly well for pruning at a much lower cost.

