# OpenReview forum: "Shaving Weights with Occam's Razor: Bayesian Sparsification for Neural Networks using the Marginal Likelihood"
_approximateinference.org/AABI/2024/Symposium — AABI 2024_

### Official Review · Reviewer_vR2c · 2024-04-18
**Interesting experimental results but the discussion of existing works is incomplete**

**Rating:** 6
**Confidence:** 4

**Review:**

Summary:

Sparsification/pruning is important to compress neural networks. However, it is not always clear to what extent we can prune a network without hurting its predictive ability. The marginal likelihood is a popular criterion for model selection in Bayesian machine learning. The authors propose to use it to sparsify Bayesian neural networks (BNN). There is a nice trick to preserve the Kronecker-factored structure of the covariance of the posterior. The authors provide experiments with promising results.

Strengths:

- The paper is overall clear and easy to follow.
- The experimental results are intriguing.

Weaknesses:

- The review of the literature covers existing works on pruning on the one hand, and existing works on BNNs on the other hand. I'm not a specialist of neural networks but I really think the authors missed all the existing works already using Bayesian approaches to prune BNNs. I think some of them lead to methods that are less scalable as the one proposed by the authors, but they should still be cited and discussed: as these papers already use criteria related to the marginal likelihood, they already take into account the predictive performance in the pruning algorithm. Ultimately, a comparison to the state-of-the-art Bayesian pruning approaches is necessary.

Recommendation:

Overall, I think the experimental results are worth being presented and discussed at the AABI 2024 Workshop track, so I will not oppose acceptation. Due to the missing references, I would however oppose publication in the conference proceedings if the paper was submitted to the archival track. I strongly encourage the authors to improve the discussion on existing works.

******

Details on the missing references:

1) optimization of the Bayesian marginal likelihood is closely related to minimization of PAC-Bayes bounds and maximization of the ELBO (aka variational inference):

Germain, P., Bach, F., Lacoste, A., & Lacoste-Julien, S. (2016). PAC-Bayesian theory meets Bayesian inference. Advances in Neural Information Processing Systems, 29.

Thus, the references should include works using ELBO and PAC-Bayes bounds for pruning. Note that the minimum description length (MDL) approach is also closely related and leads to optimization criterion very similar to the ELBO and PAC-Bayes bounds.

2) ELBO / variational inference was used for BNN pruning:

Zhao, C., Ni, B., Zhang, J., Zhao, Q., Zhang, W., & Tian, Q. (2019). Variational convolutional neural network pruning. In Proceedings of the IEEE/CVF conference on computer vision and pattern recognition (pp. 2780-2789).

Yang, Yibo, Robert Bamler, and Stephan Mandt. "Variational bayesian quantization." International Conference on Machine Learning. PMLR, 2020.

Bai, J., Song, Q., & Cheng, G. (2020). Efficient variational inference for sparse deep learning with theoretical guarantee. Advances in Neural Information Processing Systems, 33, 466-476.

It is not possible not to compare your method to Bai et al... see also the survey on NN pruning, that covers more work using the ELBO:

He, Yang, and Lingao Xiao. "Structured pruning for deep convolutional neural networks: A survey." IEEE Transactions on Pattern Analysis and Machine Intelligence (2023).

3) PAC-Bayes bounds were used for BNN pruning:

Zhou, W., Veitch, V., Austern, M., Adams, R. P., & Orbanz, P. (2018). Non-vacuous generalization bounds at the imagenet scale: a PAC-bayesian compression approach. arXiv preprint arXiv:1804.05862.

Hayou, S., He, B., & Dziugaite, G. K. (2021). Probabilistic fine-tuning of pruning masks and PAC-Bayes self-bounded learning. arXiv preprint arXiv:2110.11804.

Sakamoto, K., & Sato, I. (2022). Analyzing lottery ticket hypothesis from PAC-bayesian theory perspective. Advances in Neural Information Processing Systems, 35, 30937-30949.

I would also mention

Lotfi, S., Finzi, M., Kapoor, S., Potapczynski, A., Goldblum, M., & Wilson, A. G. (2022). Pac-bayes compression bounds so tight that they can explain generalization. Advances in Neural Information Processing Systems, 35, 31459-31473.

that uses other compression methods.

4) MDL was also discussed for deep networks:

Blier, Léonard, and Yann Ollivier. "The description length of deep learning models." Advances in Neural Information Processing Systems 31 (2018).

5) Finally, there are some theoretical works on sparse BNN that could be cited. I'm not sure scalability is the main objective of these works, so they might not have competitive algorithms to compare to, but they provide insights on why and how ELBO/PAC-Bayes can lead to approximately optimal pruning of BNN:

Chérief-Abdellatif, B. E. (2020, November). Convergence rates of variational inference in sparse deep learning. In International Conference on Machine Learning (pp. 1831-1842). PMLR.

Steffen, Maximilian F., and Mathias Trabs. "PAC-Bayes training for neural networks: sparsity and uncertainty quantification." arXiv preprint arXiv:2204.12392 (2022).

---

### Official Review · Reviewer_Jjrw · 2024-04-18

**Rating:** 6
**Confidence:** 4

**Review:**

The paper proposes to learn different precision levels of the Gaussian prior for different layers/nodes/weights to encourage different sparsity levels using ML-II and Laplace approximation. The authors also propose a new pruning criterion using the posterior mean and precision. The results indicate impressive improvements.

The paper suffers from the following weaknesses:
1. Optimizing hyper-parameters in the prior with maximizing marginal likelihood is known to have a high risk of overfitting, especially when optimizing different precision levels for different weights/nodes. This should be appropriately addressed or at least discussed.
2. The authors only considered the Gaussian prior, which is not a typical sparsity-inducing prior. Some sparsity-inducing priors, such as the horseshoe, are more appropriate.
3. Experimental setups are not clear, especially it is not clear about how the hyper-parameters of baselines are selected.

---

### Official Review · Reviewer_Ksd4 · 2024-04-21
**A comprehensive work on using Bayesian sparsification to prune Neural Networks**

**Rating:** 7
**Confidence:** 3

**Review:**

**- Summary**
The authors propose a novel pruning framework, Sparsifiability via the Marginal Likelihood (SpaM), which leverages the Bayesian marginal likelihood with sparsity-inducing priors to efficiently prune the neural network model. To address computational costs, both diagonal and KFAC Laplace approximations are tested.

**- Pros**
- The authors conducted a comprehensive study using multiple priors, criteria, network structures, and datasets. The results suggest that the proposed SpaM method can often maintain high performance even in cases of high sparsity.

**- Question**
I don't see any cons in this work, but I do have a question:
- Although Appendix E.11 briefly discusses the computational cost and provides some time metrics, I am wondering how the computational time scales as the network size increases, especially since recent BNNs typically need to use the variational method to estimate large models.

**- Conclusion**
Overall, I believe the paper presents a very promising method for pruning Bayesian neural networks. While the questions I have raised may not be essential for validating the results, addressing them could provide additional insights into the computational limitations of the SpaM.

---

### Official Review · Reviewer_cM1u · 2024-04-22

**Rating:** 6
**Confidence:** 3

**Review:**

The paper proposes to first utilize marginal likelihood to sparsify the network using a fully Bayesian principle. The true marginal likelihood is intractable, therefore the authors propose to use Laplace approximation to get an estimation of the marginal likelihood. The usage of Laplace, can later be used to provide a score for pruning, in that the diagonal of the Hessian measures the importance of a particular neuron to the output. The paper then empirically demonstrates the effectiveness of the method: It shows better performance than baseline methods at a higher level of sparsity rates. In addition, the method allows for both online pruning and post-hoc pruning and possesses uncertainty quantification capability.

Overall, I think this is a good and solid paper.

Below are some comments and questions:
1. Does standard Laplace v.s. linearized Laplace (LLLA) matter in this setting?
2. Fig.2 is nice, but it would be good if the authors could, e.g. bold the blue lines in the legend, such that readers can immediately recognize what are the methods the authors proposed or are advocating.
3. The proposed method, SpaM requires modification to the prior and the training loop, does it mean it cannot be applied on pre-trained models?

---

### Official Review · Reviewer_7o7f · 2024-04-26
**Review of Shaving Weights with Occam's Razor: Bayesian Sparsification for Neural Networks using the Marginal Likelihood**

**Rating:** 8
**Confidence:** 3

**Review:**

The paper presents a comprehensive study on neural network sparsification, and in particular sparsifiability, focusing on a Bayesian approach through the development of the Sparsifiability via the Marginal likelihood (SpaM) method. The authors introduce a novel criterion for pruning, the Optimal Posterior Damage (OPD), which leverages the computational benefits of the Laplace approximation.

Key contributions of the work include:

- A Critique of Existing Sparsification Approaches: The paper provides a critique of current methods in neural network pruning, emphasizing the limitations of existing approaches in achieving high sparsity levels without significant loss of performance. The authors argue that these methods fail to leverage the full potential of Bayesian model selection and marginal likelihood estimation.

- Introduction of SpaM and OPD: The paper introduces the SpaM framework, which implements an automatic Occam’s razor to guide the pruning process by optimizing the marginal likelihood with sparsity-inducing priors. OPD, as a novel pruning criterion derived from the Laplace approximation of the posterior Hessian, is shown to outperform traditional criteria like weight magnitude and gradient-based pruning.

On the experimental side, the authors show that:

- Existing Pruning Methods' Limitations: Conventional sparsification techniques often struggle to maintain performance at high sparsity levels, a challenge that SpaM addresses effectively. For example, in Figure 1, the authors illustrate how existing methods degrade in performance on standard datasets as sparsity increases.

- Effectiveness of SpaM and OPD: The experimental results demonstrate that SpaM, particularly when combined with OPD, leads to superior performance across various network architectures and datasets. Moreover, the increased sparsifiability of SpaM comes at no additional cost in unpruned performance (Figure 4 ).

As a concluding remark, the only weak point of the work is that the empirical validation is limited to rather small datasets. Since the method proposed by the authors seems scalable, I would advise them in a future version of the work to include experimental validation on much larger datasets (e.g. CelebAHQ) and on larger architectures (e.g. Transformers for NLP, maybe only finetuning starting from pretrained models).

---

### Meta-Review · Area_Chair_bwjr · 2024-05-26

**Recommendation:** Accept (Poster)
**Confidence:** 4

**Metareview:**

The paper studies the task of neural network sparsification from a Bayesian perspective. The authors specifically introduce a new technique based on the marginal likelihood for sparsification that takes advantage of the benefits Laplace approximations. While the reviewers agree that the authors conducted a comprehensive study that considered multiple priors, network structures, and datasets, there is a lot of work that the authors dont mention in their related works that consider other bayesian methods for pruning bayesian neural networks that should be discussed. Having said that, the approach is technically sound and reasonably novel so I agree with accepting the paper.

---

### Decision · Program_Chairs · 2024-05-27

Accept